# Sampling-free Inference for Ab-Initio Potential Energy Surface Networks

**Nicholas Gao, Stephan Günnemann**
Department of Computer Science & Munich Data Science Institute
Technical University of Munich, Germany
{n.gao,s.guennemann}@tum.de

## Abstract

Recently, it has been shown that neural networks not only approximate the ground-state wave functions of a single molecular system well but can also generalize to multiple geometries. While such generalization significantly speeds up training, each energy evaluation still requires Monte Carlo integration which limits the evaluation to a few geometries. In this work, we address the inference shortcomings by proposing the **P**otential **le**arning from **a**b-initio **Net**works (PlaNet) framework, in which we simultaneously train a surrogate model in addition to the neural wave function. At inference time, the surrogate avoids expensive Monte-Carlo integration by directly estimating the energy, accelerating the process from hours to milliseconds. In this way, we can accurately model high-resolution multi-dimensional energy surfaces for larger systems that previously were unobtainable via neural wave functions. Finally, we explore an additional inductive bias by introducing physically-motivated restricted neural wave function models. We implement such a function with several additional improvements in the new PESNet++ model. In our experimental evaluation, PlaNet accelerates inference by 7 orders of magnitude for larger molecules like ethanol while preserving accuracy. Compared to previous energy surface networks, PESNet++ reduces energy errors by up to $74\%$.

## 1 Introduction

Solving the Schrödinger equation is key to assessing a molecular system's quantum mechanical (QM) properties. As analytical solutions are unavailable, one must rely on expensive approximate solutions. Such methods are called *ab-initio* if they do not rely on empirical data. Recently, neural networks succeeded in such ab-initio calculations within the variational Monte-Carlo (VMC) framework (Carleo & Troyer, 2017). Although they lead to accurate energies, training such neural wave functions proved to be computationally intensive (Pfau et al., 2020).

To reduce the computational burden, Gao & Günnemann (2022) proposed the potential energy surface network (PESNet) to simultaneously solve many Schrödinger equations, i.e., for different spatial arrangements of the nuclei in $\mathbb{R}^3$. They use a GNN to reparametrize the wave function model based on the molecular structure. While training significantly

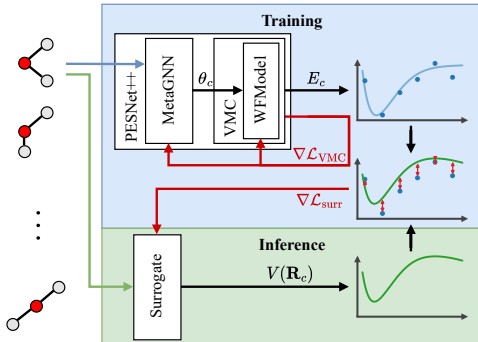

Figure 1: PlaNet framework. During training, we use the noisy energies obtained during the VMC optimization to fit our surrogate. At inference time we only query the surrogate to avoid costly numerical integration.

faster, afterward one only obtains a neural wave function that generalizes over a domain of geometries, but not the associated energy surface. Obtaining the energy of a geometry remains costly requiring a Monte-Carlo integration with complexity scaling as $O(N^4)$ in the number of electrons $N$. This high inference time prohibits many applications for neural wave functions. For instance, geometry optimization, free energy calculation, potential energy surface scans, or molecular dynamics simulations typically involve hundreds of thousands of energy evaluations (Jensen, 2010; Hoja et al., 2021).

To address the inference shortcomings, we propose the **P**otential **l**earning from **ab**-initio **Net**works (PlaNet) framework, where we utilize intermediate results from the PESNet optimization to train a surrogate graph neural network (GNN) as illustrated in Figure 1. In optimizing PESNet, one must compute approximate energy values to evaluate the loss. We propose to use these noisy intermediate energies, which otherwise would be discarded, as training labels for the surrogate. After training, the surrogate accelerates inference by directly estimating energies bypassing Monte Carlo integration.

As a second contribution, we adapt neural wave functions for closed-shell systems, i.e., systems without unpaired electrons, by introducing restricted neural wave functions. Specifically, we use doubly-occupied orbitals as inductive bias like restricted Hartree-Fock theory (Szabo & Ostlund, 2012). Together with several other architectural improvements, we present the improved PESNet++.

In our experiments, PESNet++ significantly improves energy estimates on challenging molecules such as the nitrogen dimer where PESNet++ reduces errors by $74\%$ compared to PESNet. When analyzing PlaNet we find it to accurately reproduce complex energy surfaces well within chemical accuracy across a range of systems while accelerating inference by 7 orders of magnitude for larger molecules such as ethanol. To summarize our contributions:

- **PlaNet**: an orders of magnitude faster inference method for PESNet(++), enabling exploration of higher dimensional energy surfaces at no loss in accuracy.
- **PESNet++**: an improved neural wave function for multiple geometries setting new state-of-the-art results on several energy surfaces while training only a single model.

## 2 RELATED WORK

**Neural wave functions.** Variational Monte Carlo calculations rely on expressive function forms to approximate the ground-state wave function. While early single determinants with linear combinations of basis functions without any explicit electron-electron interaction were used (Slater, 1929), later backflow (Feynman & Cohen, 1956) and Jastrow factors (Jastrow, 1955) directly introduced electron correlation effects. While the combination of both has shown success (Brown et al., 2007), Carleo & Troyer (2017) were the first to systematically exploit the expressiveness of neural networks as wave functions. While initial works targeted small systems (Kessler et al., 2021; Han et al., 2019; Choo et al., 2020), FermiNet (Pfau et al., 2020; Spencer et al., 2020), and PauliNet (Hermann et al., 2020) introduced scalable approaches. Building on their success, recent works proposed integration into diffusion Monte-Carlo (DMC) (Wilson et al., 2021; Ren et al., 2022), introduction of pseudopotentials (Li et al., 2022a), architectural improvements (Gerard et al., 2022), and extension to periodic systems (Wilson et al., 2022; Li et al., 2022b; Cassella et al., 2022). Finally, two approaches to scaling neural wave functions to multiple geometries have been explored. Scherbela et al. (2022) proposed a weight-sharing scheme across geometries to reduce the number of iterations per geometry, while Gao & Günnemann (2022) directly reparametrize the wave function with an additional neural network eliminating the need for retraining.

**Machine learning potentials.** The use of machine learning models as surrogates for quantum mechanical calculations has a rich history, e.g., one of the first force fields has been the Merck Molecular Force Field (MMFF94) (Halgren, 1996). Later, kernel methods were used (Behler, 2011; Bartók et al., 2013; Christensen et al., 2020), while graph neural networks currently obtain state-of-the-art results (Schütt et al., 2018; Gasteiger et al., 2019; 2021). Despite significant progress in the field by proving universality for certain model classes (Thomas et al., 2018; Gasteiger et al., 2021), the need for data and label quality remain limiting factors. Thus, the line between ab-initio methods and ML potentials has blurred in recent years, either by the integration of QM calculations into ML models (Qiao et al., 2020; 2021), $\Delta$-ML methods (Wengert et al., 2021), or by integrating ML models into QM calculations (Snyder et al., 2012; Kirkpatrick et al., 2021).

## 3 BACKGROUND

**Notation** For consistency with previous work, we largely follow the notation by Gao & Günnemann (2022). We use 'geometries of a molecule' to refer to different spatial arrangements of the same set of atoms in $\mathbb{R}^3$. We use $C$ to denote the number of geometries, $B$ for the number of electron configurations per geometry, $N$ for the number of electrons, and $M$ for the number of nuclei. For

electrons, we use $\mathbf{r} \in \mathbb{R}^{C \times B \times N \times 3}$ for the full tensor, $r \in \mathbb{R}^{B \times N \times 3}$ for a batch of electrons of a single geometry, $\mathbf{r} \in \mathbb{R}^{N \times 3}$ for a single set of electrons, and $r \in \mathbb{R}^3$ for a single electron. Similarly, we use $\bar{\mathbf{R}} \in \mathbb{R}^{C \times M \times 3}$, $\mathbf{R} \in \mathbb{R}^{M \times 3}$, and $R \in \mathbb{R}^3$ to denote the nuclei tensor, a nuclei geometry, and a single nucleus, respectively. We assume all coordinates in 3D space have already been transformed into the equivariant coordinate frame from Gao & Günnemann (2022) and drop the explicit transformation. We denote the local energy tensor as $\boldsymbol{E} \in \mathbb{R}^{C \times B}$ and refer to a single energy value by $E_{c,i}$. Finally, we use bracketed superscripts to index sequences, e.g., layers in a network$^{(l)}$ or training steps$^{(t)}$.

**Variational Monte Carlo.** To obtain the ground-state energy for a fixed molecular system, one has to solve the associated time-independent Schrödinger equation

$$\mathbf{H}|\psi\rangle = E|\psi\rangle. \tag{1}$$

where $\psi : \mathbb{R}^{N \times 3} \to \mathbb{R}$ is the wave function, $E \in \mathbb{R}$ is the energy and $\mathbf{H}$ is the Hamiltonian operator

$$\mathbf{H} = -\frac{1}{2} \sum_{i=1}^{3N} \nabla_i^2 + \underbrace{\sum_{i>j=1}^{N} \frac{1}{\|\boldsymbol{r}_i - \boldsymbol{r}_j\|} + \sum_{i=1}^{N}\sum_{j=1}^{M} \frac{Z_m}{\|\boldsymbol{r}_i - \boldsymbol{R}_m\|} + \sum_{m>n=1}^{M} \frac{Z_m Z_n}{\|\boldsymbol{R}_m - \boldsymbol{R}_n\|}}_{V(\mathbf{r})} \tag{2}$$

with $\nabla^2$, the Laplacian operator, and $V(\mathbf{r})$ describing the kinetic energy and potential energy, respectively. In linear algebra, Equation (1) is an eigenvalue problem where we are interested in the lowest eigenvalue $E_0$ which is also called the ground-state energy. In VMC, we accomplish this by iteratively updating the parameters $\theta$ of a trial wave function $\psi_\theta$ to approximate the ground state $\psi_0$ (McMillan, 1965). The final accuracy of VMC is mostly determined by the function class of $\psi_\theta$. Luckily, a trial function must only fulfill two requirements, first, it must be anti-symmetric to electron permutations, i.e., $\psi_\theta(\pi(\mathbf{r})) = -\operatorname{sign}(\pi)\psi_\theta(\mathbf{r})$ for all permutation $\pi$, and its integral must be finite. This enables us to use neural networks as wave functions (Carleo & Troyer, 2017). At each step, we seek to minimize the energy

$$\mathcal{L}_{\text{VMC}} = \frac{\langle \psi_\theta | \mathbf{H} | \psi_\theta \rangle}{\langle \psi_\theta | \psi_\theta \rangle} = \frac{\int \psi_\theta(\mathbf{r}) \mathbf{H} \psi_\theta(\mathbf{r}) \mathrm{d}\mathbf{r}}{\int \psi_\theta^2(\mathbf{r}) \mathrm{d}\mathbf{r}} = \frac{\int \psi_\theta^2(\mathbf{r}) \psi_\theta^{-1}(\mathbf{r}) \mathbf{H} \psi_\theta(\mathbf{r}) \mathrm{d}\mathbf{r}}{\int \psi_\theta^2(\mathbf{r}) \mathrm{d}\mathbf{r}}. \tag{3}$$

Following the standard procedure (Ceperley et al., 1977), we numerically approximate the integral via importance sampling by interpreting $\frac{\psi_\theta^2(\mathbf{r})}{\int \psi_\theta^2(\mathbf{r})\mathrm{d}\mathbf{r}}$ as a probability distribution of electron configurations $\mathbf{r}$. For each electron configuration, we then compute the so-called local energy

$$E_L(\mathbf{r}) = \psi_\theta^{-1}(\mathbf{r}) \mathbf{H} \psi_\theta(\mathbf{r}) = -\frac{1}{2} \sum_{i=1}^{N} \sum_{k=1}^{3} \left[ \frac{\partial^2 \log |\psi_\theta(\mathbf{r})|}{\partial r_{ik}^2} + \frac{\partial \log |\psi_\theta(\mathbf{r})|}{\partial r_{ik}}^2 \right] + V(\mathbf{r}). \tag{4}$$

We then use these local energies to obtain the gradients of our wave function parameters $\theta$ via

$$\nabla_\theta \mathcal{L}_{\text{VMC}} = \mathbb{E}_{\psi_\theta^2} \left[ \left( E_L(\mathbf{r}) - \mathbb{E}_{\psi_\theta^2}[E_L(\mathbf{r})] \right) \nabla_\theta \log |\psi_\theta(\mathbf{r})| \right] \tag{5}$$

where we approximate all expectations with Monte-Carlo samples obtained via Metropolis Hastings, i.e., via a Monte Carlo Markov Chain (MCMC) starting from the positions of the last iteration.

As the trial wave function $\psi_\theta$ approaches the ground state $\psi_0$, local energies approach the ground-state energy $E_0$, and their variance zero. Furthermore, our energy estimate $\mathbb{E}_{\psi_\theta^2}[E_L(\mathbf{r})]$ is lower bounded by $E_0$ because the eigenfunctions of $\mathbf{H}$ are a basis of all functions with the aforementioned properties.

While noisy estimates of the energy $\mathbb{E}_{\psi_\theta^2}[E_L(\mathbf{r})]$ are sufficient for training, at inference time, one typically has to draw $O(10^6)$ samples (Gao & Günnemann, 2022; Ren et al., 2022) to accurately model the energy for a molecular system.

**PESNet.** Since one is mostly interested in comparing energies of different geometries in chemistry, Gao & Günnemann (2022) proposed PESNet to avoid solving many Schrödinger equations independently but rather exploit the correlation between them. They accomplished this generalization by decomposing the problem into two neural networks, a wave function model (WFModel) representing the wave function $\psi_\theta$, and a problem-adapting GNN (MetaGNN) that updates the WFModels's parameters $\theta$ based on the geometry. Thus, each VMC step becomes a two-step process where one

first obtains the wave function parameters $\theta_c$ for each geometry $c \in \{1, \ldots, C\}$ via the MetaGNN and then computes the local energies and gradients with the WFModel $\psi_{\theta_c}$ via VMC. So, instead of obtaining energy estimates for the same fixed structure, during optimization, we get local energy estimates $\boldsymbol{E} \in \mathbb{R}^{C \times B}$, where $E_{c,i} = E_L(\mathbf{r}_{c,i})$, for a batch of $C$ geometries $\mathbf{R} \in \mathbb{R}^{C \times M \times 3}$.

Despite the unified training, one still needs to perform independent numerical integration per geometry to obtain energies for multiple geometries. As the cost per evaluation $O(N^4)$ grows with the number of electrons, evaluations are typically restricted to tens of geometries for small molecules due to computational constraints. Our PlaNet framework addresses these inference limitations enabling the evaluation of thousands of geometries, even for larger molecules, see Section 6.2.

## 4 PLANET – LEARNING POTENTIALS FROM NOISY INTERMEDIATE ENERGIES

To avoid the numerical integration for each geometry at inference time, we propose reusing the intermediate noisy energies $\boldsymbol{E} \in \mathbb{R}^{C \times B}$ from the optimization of PESNet(++) to train a surrogate model. This surrogate model only acts on the nuclei and outputs the energy directly, i.e., in contrast to the wave function, there is no dependence on the electrons. At inference time, we only need to query the surrogate, reducing the inference time from hours to milliseconds.

As we optimize the energy throughout the training, we must deal with the change in target labels over time. We avoid this by treating the training process as an online learning problem where we use samples only once in the order in which they were generated. This online learning biases the network towards newer lower (better) energies.

To systematically incorporate physical invariances of the energy towards the Euclidean group E(3), i.e., translation, rotation, and reflections, we use a GNN as the surrogate. Compared to traditional internal coordinate-based polynomials (Jiang & Guo, 2013; Li et al., 2013), GNNs offer straightforward generalization to larger molecules without manual feature engineering. Though, as purely distance-based GNNs are incomplete (**?**), we use the angle-encoding DimeNet$^{++}$ (Gasteiger et al., 2020).

In each training step, we train the surrogate with the current geometries $\mathbf{R}^{(t)}$ and energies $\mathbf{E}^{(t)}$ as input and target, respectively. As loss we optimize the weighted root mean squared error (RMSE)

$$\mathcal{L}_{\text{surr}}^{(t)} = \sqrt{\frac{1}{C} \sum_{c=1}^{C} \frac{\left( \hat{E}_c^{(t)} - V_\chi(\mathbf{R}_c^{(t)}) \right)^2}{\hat{\sigma}_c^{(t)}}} \tag{6}$$

where $\hat{E}_c^{(t)} = \frac{1}{B} \sum_{i=1}^{B} E_{c,i}^{(t)}$, $\hat{\sigma}_c^{(t)} = \frac{1}{B} \sqrt{\sum_{i=1}^{B} (E_{c,i}^{(t)} - \hat{E}_c^{(t)})^2}$, and $V_\chi : \mathbb{R}^{M \times 3} \to \mathbb{R}$ denotes the forward pass of DimeNet$^{++}$ with parameters $\chi$. Since we train in an online fashion, we perform $N_{\text{surr}}$ many gradient steps at each iteration $t$ to better utilize the available data.

**Accounting for label noise.** Since we obtain the energies from the VMC optimization described in Section 3, the labels are subject to noise. In fact, the observed noise is often orders of magnitude larger than the target energy differences we want to learn. To avoid the model oscillating between noisy labels, we encourage temporal data aggregation by applying an exponential moving average (EMA) to the weights $\chi$ as traditionally done in surrogate models (Gasteiger et al., 2019; 2021; Schütt et al., 2021). However, picking the decay factor for the EMA is non-trivial as large decay values may cause the model to oscillate while small decay values may cause slow convergence. To resolve this issue, we recognize that the mean absolute error (MAE) between the surrogate and the VMC estimates is lower bound by the mean absolute deviation (MAD) of the energy distribution. We exploit this to identify two regimes, an initial regime where the PlaNet training error is higher than the energy label error and one where it is lower. In the first regime, we choose a relatively high decay factor for the moving average to adapt quickly, while we shrink the decay in the second regime to encourage temporal data aggregation. To this end, we control the decay factor via

$$\gamma^{(t)} = \gamma_{\text{base}} + \gamma_{\text{high}} \mathbb{1}(\mathcal{L}_{\text{surr}}^{(t)} < \zeta D^{(t)}) \tag{7}$$

with $0 < \gamma_{\text{base}} + \gamma_{\text{high}} < 1$ and $\zeta > 1$ being hyperparameters and $\mathbb{1}$ being the indicator function. Since the true MAD $D^{(t)}$ at iteration $t$ is unknown, we must estimate it. By the central limit

theorem, we assume that the energies are approximately normally distributed with standard deviation $\hat{\sigma}^{(t)} = \frac{1}{C}\sum_{c=1}^{C}\hat{\sigma}_c^{(t)}$, thus, we approximate the MAD as $D^{(t)} \approx \sqrt{\frac{2}{\pi}}\hat{\sigma}_t$.

## 5 PESNet++ − Improving multi-geometry neural wave functions

PESNet++ builds on the recently proposed PESNet, discussed in Section 3. In this work, we adopt several improvements to enable higher-accuracy energy surfaces at a similar cost. We use restricted neural wave functions for closed-shell systems, switch from block-diagonal orbital matrices to dense ones, add a permutation invariant component to the wave function, and propose a coordinate transformation to enable larger geometric perturbations during training. Appendix A includes a complete definition of the new PESNet++ model.

**Dense orbitals.** Pfau et al. (2020); Hermann et al. (2020) and Gao & Günnemann (2022) define the wave function as a linear combination of products of determinants of spin-up and spin-down orbitals

$$\psi(\mathbf{r}) = \sum_{k=1}^{K} w_k \det \phi^{k\uparrow} \det \phi^{k\downarrow} = \sum_{k=1}^{K} w_k \det \begin{bmatrix} \phi^{k\uparrow} & 0 \\ 0 & \phi^{k\downarrow} \end{bmatrix}, \phi^{k\alpha} \in \mathbb{R}^{N^\alpha \times N^\alpha} \quad (8)$$

where $\alpha \in \{\uparrow, \downarrow\}$ indicates the spin and $\boldsymbol{w} \in \mathbb{R}^k$ are learnable weights. Equivalently, one may see the product of determinants as a determinant of an $N \times N$ block-diagonal matrix. Instead of restricting the orbital matrix to be block-diagonal, we adopt the improvements discovered by (Pfau et al., 2020) in the official FermiNet repository and define a dense orbital matrix as

$$\psi(\mathbf{r}) = \sum_{k=1}^{K} w_k \det \phi^k, \phi^k = \begin{bmatrix} \phi^{k\uparrow} \\ \phi^{k\downarrow} \end{bmatrix} \in \mathbb{R}^{N \times N}, \phi^{k\alpha} \in \mathbb{R}^{N^\alpha \times N}. \quad (9)$$

This is equivalent to picking $N$ orbital functions for each spin instead of $N^\alpha$ ones for each spin $\alpha$. Lin et al. (2021); Ren et al. (2022) and Gerard et al. (2022) have shown this to improve variational energies.

**Restricted Neural Wave Functions.** The Fermi-Dirac statistic only applies to electrons of the same spin requiring us to separate between spin-up $\uparrow$ and spin-down $\downarrow$ electrons. But, as most systems of interest, e.g., stable molecules, are closed-shell, i.e., systems without unpaired electrons, all orbitals will be doubly occupied (Szabo & Ostlund, 2012). We adopt this closed-shell formulation as inductive bias by enforcing identical orbital functions for both spins. One may see this as a neural analog to restricted Hartree-Fock (RHF), where one uses the same orbital functions for both spins, whereas previous work is related to unrestricted Hartree-Fock (UHF) with independent orbitals for both spins (Szabo & Ostlund, 2012). While UHF traditionally results in lower variational energies, we found this inductive bias to be better suited for neural wave functions, see Section 6, potentially due to its easier optimization by parameter sharing.

Speaking in symmetries, this restriction introduces invariance to the exchange of all spin-up and spin-down electrons. As both spin states are physically identical, all observables and, thus, the amplitude of the wave function stays the same under the exchange of spin states, i.e., $|\psi(\mathbf{r}^\uparrow, \mathbf{r}^\downarrow)| = |\psi(\mathbf{r}^\downarrow, \mathbf{r}^\uparrow)|$.

We implement this restriction by reformulating the interaction layers to only depend on the spin equality between two particles rather than their absolute spins. To incorporate this, we reformulate the update rules of our interaction layer to

$$\boldsymbol{h}_i^{l+1} = \sigma\left(\boldsymbol{W}_{\text{single}}^l \left[\boldsymbol{h}_i^l, \sum_{j \in \mathbb{A}^{\alpha_i}} \boldsymbol{g}_{ij}^l, \sum_{j \in \mathbb{A}^{\tilde{\alpha}_i}} \boldsymbol{g}_{ij}^l\right] + \boldsymbol{b}_{\text{single}}^l + \boldsymbol{W}_{\text{global}}^l \left[\sum_{j \in \mathbb{A}^{\alpha_i}} \boldsymbol{h}_j^l, \sum_{j \in \mathbb{A}^{\tilde{\alpha}_i}} \boldsymbol{h}_j^l\right]\right), \quad (10)$$

$$\boldsymbol{g}_{ij}^{l+1} = \sigma(\boldsymbol{W}_{\alpha_i=\alpha_j}^l \boldsymbol{g}_{ij}^t + \boldsymbol{b}_{\alpha_i=\alpha_j}^l) \quad (11)$$

where $\boldsymbol{h}_i$ and $\boldsymbol{g}_{ij}$ are single and pairwise electron features, $\alpha_i$ is the spin of the $i$-the electron, $\tilde{\alpha}$ denotes the opposing spin of $\alpha$, and $\mathbb{A}^\alpha$ is the index set of all $\alpha$ spin electrons. Subscripts $\alpha = \beta$ indicate different weights depending on whether the spins $\alpha$ and $\beta$ are identical. While Wilson et al. (2021) and Gerard et al. (2022) studied similar update functions, we are the first to explore the restricted closed-shell setting where one restricts the orbital functions to be identical between spins.

Finally, it remains to reformulate the orbital construction as

$$\phi^k = \begin{bmatrix} \phi^{k\uparrow\uparrow} & \phi^{k\uparrow\downarrow} \\ \phi^{k\downarrow\uparrow} & \phi^{k\downarrow\downarrow} \end{bmatrix}, \tag{12}$$

$$\phi_{ij}^{k\alpha\beta} = \left( (\boldsymbol{w}_i^{k,\alpha=\beta})^T \boldsymbol{h}_j^{(L_{\mathrm{WF}})} + b_i^{k,\alpha=\beta} \right) \sum_{m=1}^{M} \pi_{im}^{k\alpha=\beta} \exp(-\sigma_{im}^{k\alpha=\beta} \|r_j - R_m\|). \tag{13}$$

We initialize all $\boldsymbol{w}_i^{k,\neq}$ with zeros to ensure identical initial behavior to block-diagonal orbitals.

**Jastrow factor.** To ease optimization, we follow classical quantum mechanical methods and introduce a permutation invariant part to the wave function, the so-called *Jastrow* factor (Jastrow, 1955; Brown et al., 2007; Hermann et al., 2020). We implement the Jastrow factor $J : \mathbb{R}^{N\times D} \to \mathbb{R}$ as a permutation invariant function on the final electron embeddings $\boldsymbol{h}^{(L_{\mathrm{WF}})}$:

$$\psi(\mathbf{r}) = \exp\left( J\left( \boldsymbol{h}^{(L_{\mathrm{WF}})} \right) \right) \sum_{k=1}^{K} w_k \det \phi^k, \qquad\qquad J(\boldsymbol{h}) = \sum_{i=1}^{N} \mathrm{MLP}(\boldsymbol{h}_i). \tag{14}$$

**Avoiding dead neurons.** In recent years, the flow of information in neural networks has been heavily studied (Brock et al., 2022; 2021; Hendrycks & Gimpel, 2020). While such works have shown great success in classical deep learning tasks such as computer vision, they have not yet been applied to neural wave functions. We implement several such improvements: careful parameter initialization, rescaling after activation functions (Brock et al., 2022), and the SiLU activation function (Hendrycks & Gimpel, 2020). As all features are learned based on Euclidean distances, we still use $\tanh$ after the first layer to limit the magnitude of feature embeddings. We observed significant improvements in performance and trace it back to a significantly reduced number of dead neurons, i.e., the number of neurons whose value is independent of the input. We illustrate these observations in Appendix B.

**Coordinate transform.** In machine learning, one typically draws a batch of samples independently from some data distribution. Unfortunately, we cannot do the same for molecular geometries. Recall from Section 3, as we use an MCMC to sample the electron positions, we have to keep track of the last electron positions for each geometry to sample new ones. Thus, instead of i.i.d. sampling, Gao & Günnemann (2022) obtain the next batch of geometries by applying small perturbations to the previous one. While performing sufficiently well for low-dimensional energy surfaces, such small changes do not scale to higher-dimensional problems where the search space grows exponentially. To close the gap to the i.i.d. setting, we increase the magnitude of perturbations. However, this comes with the danger of decorrelating the electron positions from the nuclei. We minimize such decorrelation effects by transforming each electron in accord with its closest nucleus. This displacement avoids electron configurations from reaching dead regions of the probability distribution by guaranteeing that the distance between an electron and its closest nucleus may only decrease. Formally, if $\mathbf{R}' \in \mathbb{R}^{M\times 3}$ are the new nuclei positions we obtain the new electron positions $\mathbf{r}'$ via $\lambda : \mathbb{R}^3 \times \mathbb{R}^{M\times 3} \times \mathbb{R}^{M\times 3} \to \mathbb{R}^3$

$$\lambda(\boldsymbol{r}|\mathbf{R}', \mathbf{R}) = \boldsymbol{r} + (\boldsymbol{R}_i' - \boldsymbol{R}_i), \tag{15}$$
$$i = \arg\min_m \|\boldsymbol{R}_m - \boldsymbol{r}\|_2.$$

LIMITATIONS

Firstly, while PlaNet enables high-resolution multi-dimensional energy surfaces, scaling to higher-dimensional energy surfaces remains a challenge for all computational methods. Due to the curse of dimensionality, high dimensional energy surfaces will only be sparsely sampled, e.g., in Appendix G.5 we find PlaNet exceeding the threshold of chemical accuracy once modeling the full six-dimensional energy surface of $H_2$-HF. To lessen this issue, one could look into systematically reusing previous labels, extending the training time for higher dimensional energy surfaces, or adopting equivariant surrogate architectures as these have shown to be more sample efficient (Batzner et al., 2022).

Secondly, DimeNet$^{++}$ cannot distinguish some molecules. As this work seeks to present the general PlaNet framework, the choice of surrogate architecture is not an essential part of this work. We encourage the adoption of universal models in later works (Thomas et al., 2018; Gasteiger et al., 2021).

Table 1: MAE↓ in $\mathrm{m}E_\mathrm{h}$ of VMC energy surfaces compared to experimental results on N$_2$ (Le Roy et al., 2006). PESNet++ reduces the error by 74 %.

|  | MAE ↓ |
|---|---|
| PESNet | 5.36 |
| + SiLU | 4.90 |
| + zero bias | 4.23 |
| + Jastrow | 3.29 |
| + Dense orbitals | 2.60 |
| + Restricted | 1.39 |

Table 2: MAE↓ in $\mathrm{m}E_\mathrm{h}$ between VMC and PlaNet over several energy surfaces. SE indicates the standard error of VMC energies in $\mathrm{m}E_\mathrm{h}$. Brackets indicate the standard deviation at the last digit(s) across 5 trainings.

|  | MAE ↓ | SE |
|---|---|---|
| H$_4$ | 0.0089(5) | 0.004 |
| Li$_2$ | 0.051(11) | 0.019 |
| H$_{10}$ | 0.152(25) | 0.018 |
| N$_2$ | 0.26(5) | 0.186 |
| H$_2$-HF | 0.324(14) | 0.14 |
| C$_2$H$_5$OH | 0.298(26) | 0.33 |

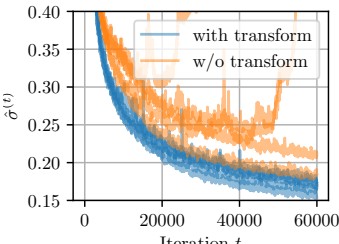

Figure 2: Energy standard deviation $\hat{\sigma}^{(t)}$↓ during optimization with and without coordinate transform. For each setting 5 runs are plotted.

Lastly, the restricted neural wave function formulation is only suitable for molecules where the ground state is a singlet state. For molecules with multiple unpaired electrons, for instance the O$_2$ ground-state, or atomic systems, restricted neural wave functions cannot properly describe the ground state and, thus, lead to higher energies.

## 6 EXPERIMENTS

We split our experiments into two parts. Firstly, we present exhaustive ablation studies on the contributions from Sections 4 and 5. Secondly, we analyze PlaNet's scaling by applying it to several potential energy surfaces, including the challenging nitrogen dimer, a two-dimensional energy surface, and a high-resolution energy surface of the larger ethanol molecule.

We denote the sampling-free inference energy estimates as PlaNet and the VMC energies by PES-Net++. Note that VMC energies are guaranteed to be upper bounds to the true energy, while no such a bound exists for the PlaNet energies. When evaluating PlaNet energies, PESNet++ should be seen target since PlaNet should surrogate the VMC integration. Appendix D details the setup, and Appendix C the optimization. Timings are given in Nvidia A100 GPU hours.

### 6.1 ABLATION STUDIES

**PESNet++.** Firstly, we evaluate the proposed architectural changes of PESNet++ (Section 5) on the nitrogen molecule in Table 1. We picked the nitrogen molecule as it is known for strong electron correlation effects that result in relatively high errors for most computational methods (Gdanitz, 1998; Pfau et al., 2020; Gao & Günnemann, 2022). Evidently, all improvements gradually lower the MAE to the experimental results leading to a total error reduction of 74 %. Interestingly, restricting the wave function to doubly occupied orbitals results in the largest improvements, both in percentage (46 %) and absolute value ($1.21\,\mathrm{m}E_\mathrm{h}$). See Appendix H for a training dynamics comparison.

**Coordinate transform.** To identify whether the in Section 5 proposed coordinate transform benefits training on multi-dimensional energy surfaces, we train multiple PESNet++ with and without the coordinate transform, on the full six-dimensional energy surface of H$_2$-HF and observe the convergence of the standard deviation of the energy $\hat{\sigma}^{(t)}$. As the wave functions converge we expect $\hat{\sigma}^{(t)}$ to approach 0. Figure 2 shows the convergence of several PESNet++ with and without the proposed coordinate transform. We find our transformation aiding in consistency and quality of convergence. Noticeably, without our coordinate transforms 3 out of 5 runs diverge during training. Compared to the converged runs, final standard deviations are on average 17 % lower, reducing the number of samples to obtain the same statistical error by 31 %.

**PlaNet energies.** Lastly, we analyze PlaNet's reconstruction of VMC energies across a variety of energy surfaces, see Appendix J for the list of used geometries. The low errors in Table 2 suggest a very close fit between the PlaNet and VMC energies as all deltas are significantly below the threshold for chemical accuracy of $1.6\,\mathrm{m}E_\mathrm{h}$. Further, for larger systems, the fit is indistinguishable from

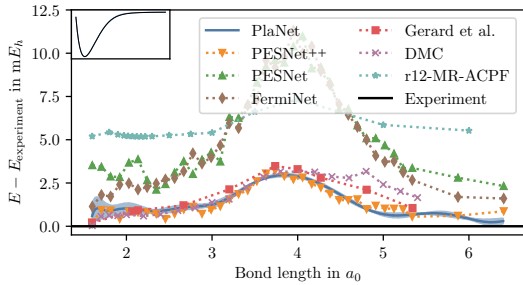

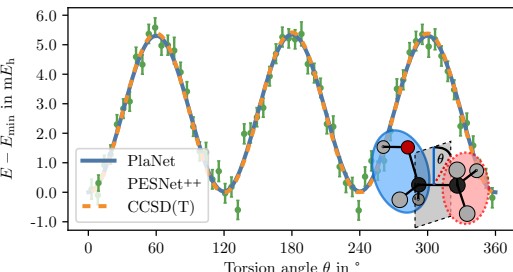

Figure 3: Potential energy curve of $N_2$ (Pfau et al., 2020; Gao & Günnemann, 2022; Le Roy et al., 2006; Gerard et al., 2022; Ren et al., 2022).

Figure 4: Potential energy curve of the $CH_3$ torsion angle of Trans-ethanol (Nandi et al., 2022). Error bars indicate the standard error.

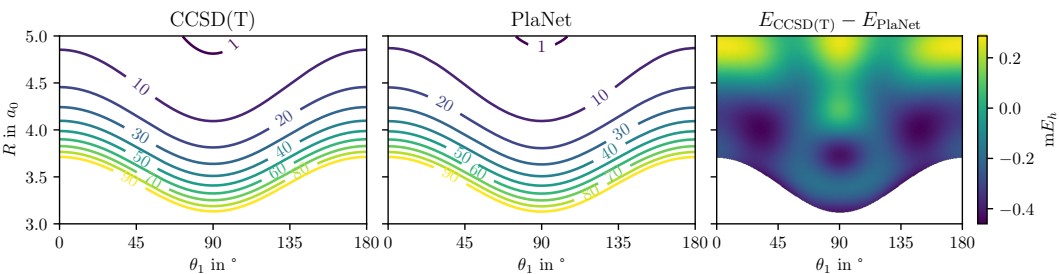

Figure 5: Two-dimensional slice of the potential energy surface of $H_2$-HF. The remaining four dimensions are fixed at the equilibrium structure. Energies are zeroed at the equilibrium structure and in $mE_h$. PlaNet agrees well ($\approx 0.28(9)\, mE_h$) with the CCSD(T)-based fit (Yang et al., 2018).

the 'ground-truth' VMC energies as the VMC error per geometry increases. Appendix I presents additional ablation studies on PlaNet hyperparameters and Appendix E an analysis of relative errors.

## 6.2 POTENTIAL ENERGY SURFACES

Evaluating *ab-initio* methods remains a complicated issue as true energy values are rarely known. Furthermore, the threshold for chemical accuracy at $1\,\mathrm{kcal\,mol}^{-1} \approx 1.6\,mE_h$ is often relatively small compared to the absolute magnitude of energies. However, as energy differences in chemistry are much more important than their absolute value, we follow the standard procedure (Pfau et al., 2020; Hermann et al., 2020; Gerard et al., 2022; Gao & Günnemann, 2022) and compare relative energy surfaces. As references, we use CCSD(T) or experimental results. Additional experiments on the lithium dimer (G.1), the hydrogen chain (G.2), cyclobutadiene (G.3), different ethanol states (G.4), and higher-dimensional energy surfaces of $H_2$-HF (G.5) are available in Appendix G.

**Heavily correlated systems.** The nitrogen dimer is known to be a challenging system due to significant electron correlation effects (Le Roy et al., 2006), causing even the 'gold-standard' CCSD(T) method to fail (Lyakh et al., 2012). Nonetheless, FermiNet presented very accurate results comparable to the factorially scaling r12-MR-ACPF method (Pfau et al., 2020; Gdanitz, 1998). The comparison in Figure 3 shows that PESNet++ estimates the lowest variational energies. As for the relative error $\Delta E_{\max} - \Delta E_{\min}$, PESNet++ outperforms Gerard et al. (2022)'s FermiNet extension and expensive DMC methods with a relative error of $3\,mE_h$ compared to $3.2\,mE_h$ for DMC (Ren et al., 2022) and Gerard et al. (2022), despite training only a single model for a fraction of the time. Finally, PlaNet's equilibrium distance of $2.0747\,a_0$ is very close to the experimental results at $2.0743\,a_0$ (Le Roy et al., 2006). For details on finding the equilibrium geometries see Appendix K.

**Scaling to larger systems.** To explore the generalization towards larger systems, we investigate the potential energy surface of Trans-ethanol along the CH3 torsion angle. In Figure 4, we compare PESNet++ and PlaNet estimates with accurate CCSD(T) results from Nandi et al. (2022). PlaNet agrees with the 99% confidence interval of all PESNet++ energies. Further, PlaNet almost perfectly

recovers the CCSD(T) results with a maximum disagreement of $0.13 \, mE_h$ while only requiring $\approx 0.11 \, ms$ per inference after training compared to the $\approx 2.7 \, h$ per inference required for VMC integration. Additionally, the PlaNet energies are more robust and less noisy than the VMC energies obtained via numerical integration. While this discrepancy can be closed by increasing the number of samples during Monte-Carlo integration, reducing the VMC error bars by an order magnitude requires $\approx 270 \, h$ per geometry. This consistency is an encouraging sign that PlaNet-based interpolation has significant advantages in multi-geometry VMC over traditional post hoc interpolation methods (Nandi et al., 2022; Yang et al., 2018; Jiang & Guo, 2013). In Appendix G.4, we explore the same CH3 torsion angle energy surfaces for different ethanol states.

**Two-dimensional energy surface.** Finally, we are interested in PlaNet's scaling to multi-dimensional energy surfaces. We chose a two-dimensional slice of the $H_2$-HF interaction energy surface with CCSD(T) reference calculations (Yang et al., 2018) as described in Appendix F. Figure 5 depicts a comparison between the two energy surfaces. PlaNet accurately reproduces this high-resolution two-dimensional energy surface with an MAE of $\approx 0.28(9) \, mE_h$, leading to a T-shaped equilibrium structure at $R = 5.34 \, a_0$ and $\theta_1 = 90°$ consistent with previous results (Bernholdt et al., 1986; Yang et al., 2018; Guillon et al., 2008). Note that such a high-resolution energy surface would not have been obtainable with previous neural wave function methods, see Appendix 6.3. In Appendix G.5, we analyze the fitting of PlaNet to higher-dimensional slices of the energy surface.

## 6.3 TRAINING AND INFERENCE TIMES

In the previous experiments, we have shown that PlaNet approximates the underlying Monte Carlo integration well with little to no additional errors. In this section, we put the training and inference times of PlaNet's surrogate in perspective to the standard VMC procedure. In Table 3, we list the required times for training PESNet++ (VMC training), performing Monte Carlo integration (VMC inference), training the surrogate, and performing surrogate inference. While the training times refer to training the model to a whole energy surface, the inference times are given per geometry. Note that we exclude pretraining and initial MCMC thermalization in training since these contribute only a small frac-

Table 3: VMC and surrogate training and inference times per system. Inferences times are given per geometry. The training of the surrogate accounts for $<1\%$ of total training times while accelerating inference by 6 to 9 orders of magnitude.

| | VMC | | Surrogate | |
|---|---|---|---|---|
| | Training | Inference | Training | Inference |
| $H_4$ | 32 h | 9.4 min | 20 min | 4.7 μs |
| $Li_2$ | 40 h | 13.6 min | 18.5 min | 0.75 μs |
| $H_{10}$ | 81 h | 26.8 min | 47 min | 188 μs |
| $H_2$-HF | 107 h | 38.4 min | 20 min | 4.7 μs |
| $N_2$ | 123 h | 52.7 min | 18.5 min | 0.75 μs |
| $C_2H_5OH$ | 478 h | 2.7 h | 43 min | 118 μs |

tion of the total time. Evidently, training the surrogate is equivalent to between $<1$ and 3 VMC inferences worth of time, while enabling 6 to 9 orders of magnitude faster inference. Further, surrogate training accounts for $<1\%$ of the total training time. These time advantages are growing with system size, e.g., for ethanol the surrogate training accounts for $<0.2\%$ of the total training time.

## 7 CONCLUSION

We presented the PlaNet framework for sampling-free inference of potential energy surface networks by training an additional surrogate on intermediate noisy variables from the VMC optimization. We addressed the changing energies during training by viewing the problem as an online learning task. Moreover, we identified two training regimes depending on PlaNet's accuracy and energy distribution. These regimes enabled us to perform efficient temporal data aggregation, reducing the impact of noisy labels. In addition, we explored doubly-occupied orbitals for neural wave functions by introducing PESNet++. In our evaluation, we observed that: (1) PESNet++ results in new state-of-the-art VMC results on several molecules, and (2) PlaNet suffers no loss in accuracy while decreasing inference times from hours to milliseconds. Thanks to these improvements in inference and accuracy, accurate high-resolution multi-dimensional energy surfaces with neural wave functions are obtainable. These are promising indicators for the future application of neural wave function methods for tasks that typically require hundreds of thousands or millions of energy evaluations, like analyzing multi-dimensional energy surfaces or molecular dynamics simulations.

ACKNOWLEDGEMENTS

We thank Daniel Zügner, Filippo Guerranti, and Jan Schuchardt for their invaluable feedback on the manuscript, and Marten Lienen for our discussion on visualizing torsion angles.

Funded by the Federal Ministry of Education and Research (BMBF) and the Free State of Bavaria under the Excellence Strategy of the Federal Government and the Länder.

REPRODUCIBILITY

Our source is publicly available [1] licensed under the Hippocratic license (Ehmke, 2019). Further, Appendix A details the PESNet++ architecture, Appendix C the optimization algorithm, and Appendix D the rest of the experimental setup including all hyperparameters.

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

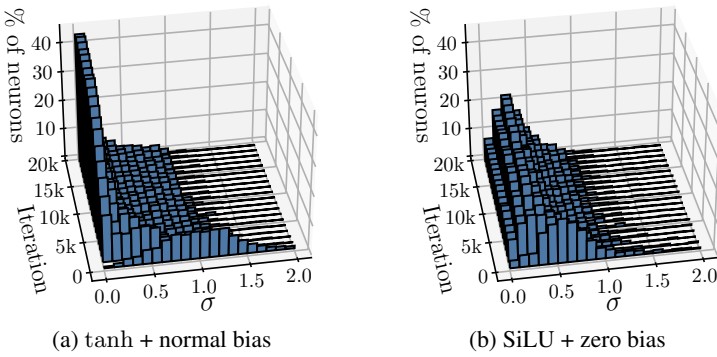

(a) $\tanh$ + normal bias
(b) SiLU + zero bias

Figure 6: Distribution of standard deviation of neurons during training.

## A  PESNET++ ARCHITECTURE

PESNet++ consists of three key ingredients, the MetaGNN, the equivariant coordinate system, and the wave function model (WFModel). The WFModel is used within the VMC framework to generate gradients and find the ground-state wave function. The MetaGNN's goal is to adapt the WFModel to the geometry at hand and the equivariant coordinate system enforces the physical symmetries of the energy. We directly adopt the MetaGNN and the equivariant coordinate system from Gao & Günnemann (2022) and would like to redirect the reader to the original work for more information. All architectural improvements in PESNet++ affect the WFModel. The WFModel acting on an electron configuration $\mathbf{r}$ first constructs single-electron $\boldsymbol{h}_i^{(1)}$ and pair-wise $\boldsymbol{g}_{ij}^{(1)}$ features in a permutation equivariant way. We adopt the same construction as in (Gao & Günnemann, 2022):

$$\boldsymbol{h}_1^{(1)} = \sum_{m=1}^{M} \text{MLP}\left(\boldsymbol{W}\left[(\boldsymbol{r}_i - \boldsymbol{R}_i)\,E, \|\boldsymbol{r}_i - \boldsymbol{R}_i\|\right] + z_m\right), \tag{16}$$

$$\boldsymbol{g}_{ij}^{(1)} = \left((\boldsymbol{r}_i - \boldsymbol{r}_j)\,E, \|\boldsymbol{r}_i - \boldsymbol{r}_j\|\right) \tag{17}$$

where $E \in \mathbb{R}^{3 \times 3}$ is our equivariant coordinate system and $z_m$ are nuclei embeddings outputted by the MetaGNN. These features are then iteratively updated with our new update rules from Equation (10) and Equation (11). After $L_{\text{WF}}$ many layers, we use the final electron embeddings $h_i^{(L_{\text{WF}})}$ to construct the orbital matrices $\phi$ with Equation (13). Finally, we compute the final amplitude with the Jastrow factor with Equation 14.

## B  DEAD NEURON ANALYSIS

To illustrate the impact of $\tanh$ as an activation function and the normally distributed biases, Figure 6a plots the standard deviation of PESNet's neurons throughout training. One may see that the number of dead neurons increases during training to $>40\,\%$, reducing the effective network size. Our improvements reduce the number of dead neurons to $<10\,\%$ as seen in Figure 6b. Note that we still keep one $\tanh$ activation function in the embedding block to limit the magnitude of our embeddings if distances increase.

## C  OPTIMIZATION

Each PlaNet optimization step consists of two parts, the first is a VMC optimization step with the additional coordinate transform, and the second is the optimization of the surrogate.

In each VMC step, we first perturb the geometry from the previous iteration followed by the coordinate transformation from Equation (15). Next, we sample for each geometry $c$ new electron positions $\boldsymbol{r}_c$ via Metropolis-Hastings and calculate the local energy $E_{c,i}$ for each electron configuration $\mathbf{r}_{c,i}$ (McMillan, 1965). Given these energies, we compute the gradients and construct our updates via natural gradient descent with the conjugate gradient (CG)-method (Neuscamman et al., 2012). For

---

**Algorithm 1** $t + 1$th optimization step

---

**Input:** $\mathbf{R}^{(t)}, \mathbf{r}^{(t)}, \Theta^{(t)}, \chi^{(t)}$
**Output:** $\mathbf{R}, \mathbf{r}, \Theta, \chi$
   $\mathbf{R} \sim \rho(\mathbf{R}|\mathbf{R}^{(t)})$
   $\mathbf{r}' = \lambda(\mathbf{r}^{(t)}|\mathbf{R}, \mathbf{R}^{(t)})$           $\triangleright$ Equation (15)
   **for** $c \in \{1, \ldots, C\}$ **do**
      $r_c \sim \psi^2_{\theta_c^{(t)}}(r_c')$           $\triangleright$ MCMC
      $E_{c,i} = \psi_{\theta_c^{(t)}}(\mathbf{r}_{c,i})^{-1} \mathbf{H} \psi_{\theta_c^{(t)}}(\mathbf{r}_{c,i})$      $\triangleright$ (Pfau et al., 2020; Hermann et al., 2020)
      $\delta_{c,i} = E_{c,i} - (\frac{1}{B} \sum_{i=1}^{B} E_{c,i})$
   **end for**
   $\nabla_{\Theta^{(t)}} \mathcal{L} = \mathbb{E}_{c,i} [\delta_{c,i} \nabla_{\Theta^{(t)}} \log |\psi_{\Theta^{(t)}}(\mathbf{r}_{c,i})|]$
   $\Theta = \Theta^{(t)} - \eta \boldsymbol{F}^{-1} \nabla_{\Theta^{(t)}} \mathcal{L}$        $\triangleright$ CG-method (Gao & Günnemann, 2022)

   **for** $i \in \{1, \ldots, N_{\text{surr}}\}$ **do**
      $\chi' \leftarrow$ AdamW step on $\mathcal{L}_{\text{surr}}$           $\triangleright$ Equation (6)
   **end for**
   Compute $\gamma$           $\triangleright$ Equation (7)
   $\chi = \gamma \chi^{(t)} + (1 - \gamma) \chi'$

---

further details on the VMC step, we would like to refer the reader to Pfau et al. (2020) and Gao & Günnemann (2022). Next, we fit our surrogate model over $N_{\text{surr}}$ many steps to the local energies and, finally, temporally average the surrogate parameters via the moving average described in Section 4. The complete optimization algorithm is given in Algorithm 1. Note that we dropped the dependence on the $t + 1$ step and explicitly wrote down the loop over all geometries. Though, in practice one implements those as an additional batch dimension and performs these operations in parallel. To reduce the dependence on the current batch, we smooth $\mathcal{L}_{\text{surr}}^{(t)}$ and $D^{(t)}$ when evaluating Equation (7) with EMAs.

## D    EXPERIMENTAL SETUP

All default hyperparameters are reported in Table 4. To avoid exhaustive hyperparameter searches, we use the default hyperparameters for PESNet (Gao & Günnemann, 2022) and DimeNet$^{++}$ (Gasteiger et al., 2020). Exceptions are the nitrogen dimer for which we increased the number of determinants to 32, the $H_2$-HF energy surfaces for which we increased the determinants to 32, and the number of geometry random walkers to 64. In contrast to PESNet (Gao & Günnemann, 2022), we do not deploy any early stopping criteria for the CG-method as we found the convergence to benefit slightly if we always do the full 100 steps. Note that this has barely any impact on the optimization time as the early stopping criteria were rarely met.

We implemented PlaNet and PESNet++ on top of the official JAX (Bradbury et al., 2018) implementation of PESNet (Gao & Günnemann, 2022) which is distributed under the Hippocratic license (Ehmke, 2019). We ran all experiments on a machine with 16 AMD EPYC 7742 cores and a single Nvidia A100 GPU.

## E    RELATIVE PLANET ERRORS

While we analyzed the quality of the potential fit in the main body, the MAE is not a perfect metric, as one is often interested in relative energies rather than their absolute value. To account for this, we present extended results in Table 5 where we also list the relative MAE and the MAD at the last iteration. We define the relative MAE as

$$\text{rel. MAE} = \frac{1}{N} \sum_{i=1}^{N} |(E_i' - \hat{E}') - (E_i - \hat{E}_i)| \tag{18}$$

Table 4: Default hyperparameters.

|  | Parameter | Value |
|---|---|---|
| Optimization | Local energy clipping | 5.0 |
| Optimization | Batch size | 4096 |
| Optimization | Iterations | 60000 |
| Optimization | #Geometry random walker | 16 |
| Optimization | Learning rate $\eta$ | $\frac{0.1}{(1+t/1000)}$ |
| Natural Gradient | Damping | $10^{-4}\sigma^{(t)}$ |
| Natural Gradient | CG steps | 100 |
| WFModel | Nuclei embedding dim | 64 |
| WFModel | Single-stream width | 256 |
| WFModel | Double-stream width | 32 |
| WFModel | #Update layers | 4 |
| WFModel | #Determinants | 16 |
| WFModel | #Jastrow layers | 3 |
| MetaGNN | #Message passings | 2 |
| MetaGNN | Embedding dim | 64 |
| MetaGNN | Message dim | 32 |
| MetaGNN | $N_{\text{SBF}}$ | 7 |
| MetaGNN | $N_{\text{RBF}}$ | 6 |
| MetaGNN | MLP depth | 2 |
| MCMC | Init proposal step size | 0.02 |
| MCMC | Steps between updates | 40 |
| Pretraining | Iterations | 2000 |
| Pretraining | Learning rate | 0.003 |
| Pretraining | Method | RHF |
| Pretraining | Basis set | STO-6G |
| Evaluation | #Samples | $10^6$ |
| Evaluation | MCMC Steps | 400 |
| DimeNet$^{++}$ | Cutoff $r_c$ | $10.0a_0$ |
| DimeNet$^{++}$ | Basis embedding size | 8 |
| DimeNet$^{++}$ | Interaction embedding size | 64 |
| DimeNet$^{++}$ | Out embedding size | 256 |
| DimeNet$^{++}$ | #Layer after skip | 2 |
| DimeNet$^{++}$ | #Layer before skip | 1 |
| DimeNet$^{++}$ | #Blocks | 4 |
| DimeNet$^{++}$ | #Layer out | 3 |
| DimeNet$^{++}$ | $N_{\text{RBF}}$ | 6 |
| DimeNet$^{++}$ | $N_{\text{SBF}}$ | 7 |
| PlaNet Optimization | $\gamma_{\text{base}}$ | 0.99 |
| PlaNet Optimization | $\gamma_{\text{high}}$ | 0.0099 |
| PlaNet Optimization | Learning rate | $\frac{10^{-4}}{(1+t/10000)}$ |
| PlaNet Optimization | $N_{\text{surr}}$ | 5 |
| PlaNet Optimization | $\zeta$ | 1.05 |
| PlaNet Optimization | Optimizer | AdamW |
| PlaNet Optimization | $\mathcal{L}_{\text{surr}}^{(t)}$ and $D^{(t)}$ EMA decay | 0.999 |

Table 5: Extended version of Table 2. MAE↓ in $mE_h$ between VMC and PlaNet absolute and relative energies. 'SE' indicates the standard error of the mean for VMC target energies. The 'MAD' row refers to the MAD of the energy distribution at the end of training. Numbers in brackets indicate the standard deviation at the last digit(s) across 5 trainings.

|  | $H_4^+$ | $H_4$ | $Li_2$ | $H_{10}$ | $N_2$ | $H_2$-HF | $C_2H_5OH$ |
|---|---|---|---|---|---|---|---|
| MAD | 0.43 | 0.3 | 1.5 | 1.4 | 9.0 | 18.8 | 15.0 |
| SE | 0.004 | 0.004 | 0.019 | 0.018 | 0.186 | 0.14 | 0.33 |
| abs. MAE↓ | 0.0110(4) | 0.0089(5) | 0.051(11) | 0.152(25) | 0.26(5) | 0.324(14) | 0.298(26) |
| rel. MAE↓ | 0.0101(4) | 0.00442(22) | 0.049(11) | 0.13(4) | 0.25(32) | 0.219(32) | 0.274(5) |

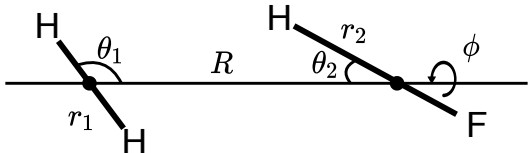

Figure 7: Internal coordinate description of the $H_2$-HF system.

where $E_i$ and $E_i'$ refer to the VMC and the PlaNet estimate of the energy of the $i$th configuration, and $\hat{E}$ and $\hat{E}'$ to their respective means.

An interesting result is the comparatively low relative error for systems with heavy nuclei, i.e., non-hydrogen nuclei. We suspect due to the exponential moving average, PlaNet is biased towards higher energies if the VMC energies converge slowly.

## F  $H_2$-HF SYSTEM

Figure 7 describes the internal coordinates of the $H_2$-HF system. We follow (Yang et al., 2018) in the definition of the equilibrium structure and set it to $r_1 = 1.4051$, $r_2 = 1.73496$, $\phi = 0$, $\theta_1 = 90$, $\theta_2 = 180$ and $R = 5.4$. For the two-dimensional slices, we fix all but $\theta_1$ and $R$ at the equilibrium structure.

## G  ADDITIONAL ENERGY SURFACES

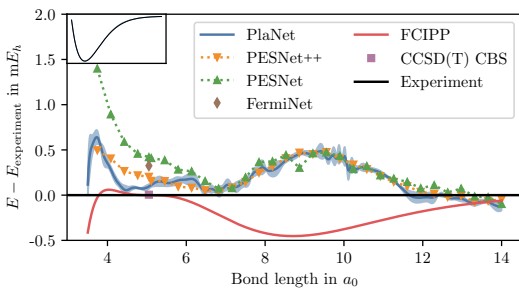

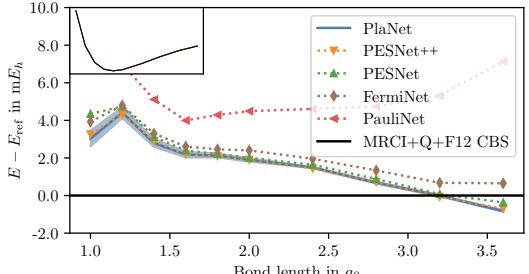

Figure 8: Potential energy curve of $Li_2$ (Pfau et al., 2020; Maniero & Acioli, 2005). Lightly shaded regions indicate standard deviation across 5 surrogate fits. The inset shows the absolute energy surface. Compared to PESNet, PESNet++ reduces the maximum error by $0.9 \, mE_h$.

Figure 9: Potential energy curve of the hydrogen chain (Pfau et al., 2020; Hermann et al., 2020; Gao & Günnemann, 2022; Motta et al., 2017). Lightly shaded regions represent the standard deviation across 5 surrogate fits. PESNet++ improves upon PESNet by $0.27 \, mE_h$.

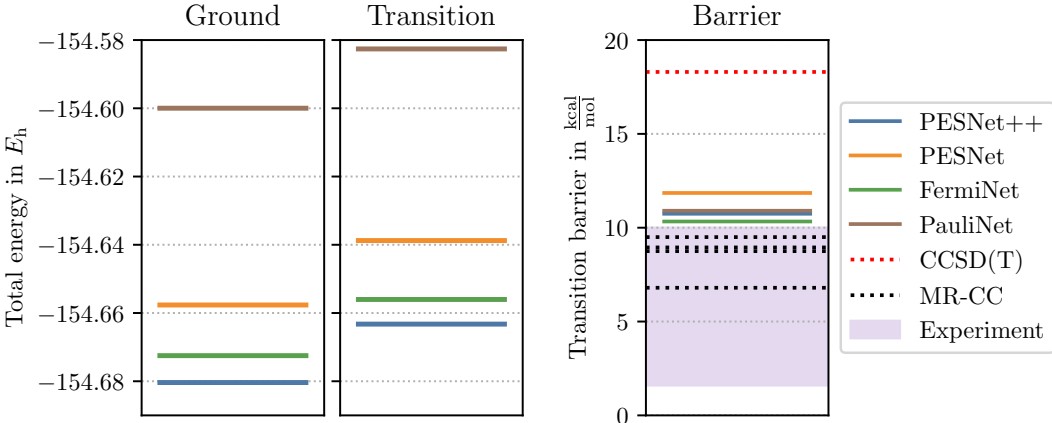

Figure 10: Comparison of PESNet++, PESNet, FermiNet, and PauliNet on estimating the transition barrier of cyclobutadiene. The first two plots show the total energy of the ground and transition state while the right plot shows estimates of the transition barrier. All neural wave function methods estimate similar transition barriers at the upper end of the experimental range. In terms of total energy, PESNet++ outperforms PauliNet by $80.5\,\mathrm{m}E_\mathrm{h}$, PESNet by $23.6\,\mathrm{m}E_\mathrm{h}$ and FermiNet by $7.6\,\mathrm{m}E_\mathrm{h}$.

### G.1 LITHIUM DIMER

A typical task in chemistry is identifying equilibrium structures. To quantify PlaNet's suitability for such tasks we apply it to the lithium dimer. One can see in Figure 8 that PlaNet agrees well with experimental results and PESNet++. Further, PlaNet is virtually indistinguishable from the VMC estimates. When searching for the equilibrium structure with PlaNet, we find the minimum at $5.041\,\mathrm{a}_0$, i.e., only $0.01\,\mathrm{a}_0$ apart from the experimental results (Maniero & Acioli, 2005). PESNet++ reduces the maximum error from $1.4\,\mathrm{m}E_\mathrm{h}$ to $0.5\,\mathrm{m}E_\mathrm{h}$ ($64\,\%$).

### G.2 HYDROGEN CHAIN

The hydrogen chain is a classic benchmark geometry with a rich history (Motta et al., 2017). Thus, we are interested in how our PlaNet and PESNet++ compare to other neural wave functions (Pfau et al., 2020; Hermann et al., 2020; Gao & Günnemann, 2022). The potential energy curve is plotted in Figure 9. We find our PESNet++ to produce the, to date, lowest energies with an improvement of $\approx 0.3\,\mathrm{m}E_\mathrm{h}$ over PESNet and $\approx 0.6\,\mathrm{m}E_\mathrm{h}$ over FermiNet. Further, the PlaNet energies only differ by $0.12\,\mathrm{m}E_\mathrm{h}$ from the VMC estimates.

### G.3 TRANSITION BARRIER OF CYCLOBUTADIENE

To compare the architectural improvements of PESNet++ to existing neural wave functions, we report total energies and transition barrier estimates in Figure 10. As in FermiNet (Spencer et al., 2020) and PESNet (Gao & Günnemann, 2022), we increased the hidden dimension of the single electron features to 512 and use 32 instead of 16 determinants. We resolved the convergence difficulties reported by Gao & Günnemann (2022) by training on the continuous energy surface obtained by linearly interpolating between the ground and transition state instead of only training on the ground and transition state. From the results in Figure 10, it is apparent that PESNet++ improves variational energies significantly by setting new state-of-the-art variational energies that are on average $7.6\,\mathrm{m}E_\mathrm{h}$ better than FermiNet's. Though, it should be noted that compared to PESNet more compute resources have been used due to the additional cost of the dense orbitals and restricted neural wave function, see Appendix H.

### G.4 DIFFERENT ETHANOL STATES

In the main body, we covered the $CH_3$ torsion angle of Trans-ethanol which is in perfect agreement with the CCSD(T) reference (Nandi et al., 2022) calculations. Here, we additionally explore the

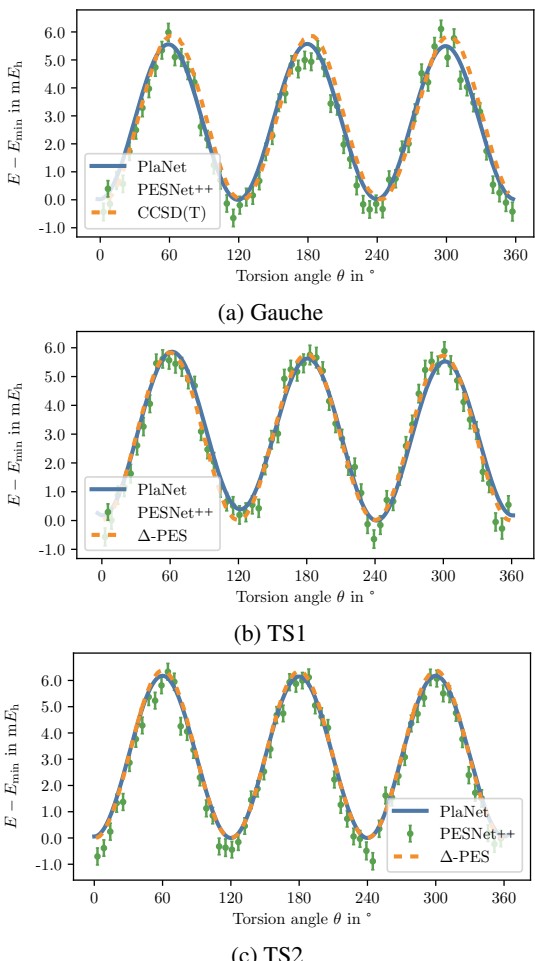

(a) Gauche

(b) TS1

(c) TS2

Figure 11: Potential energy curves of the $CH_3$ torsion angle of ethanol for different OH torsion angle arrangements.

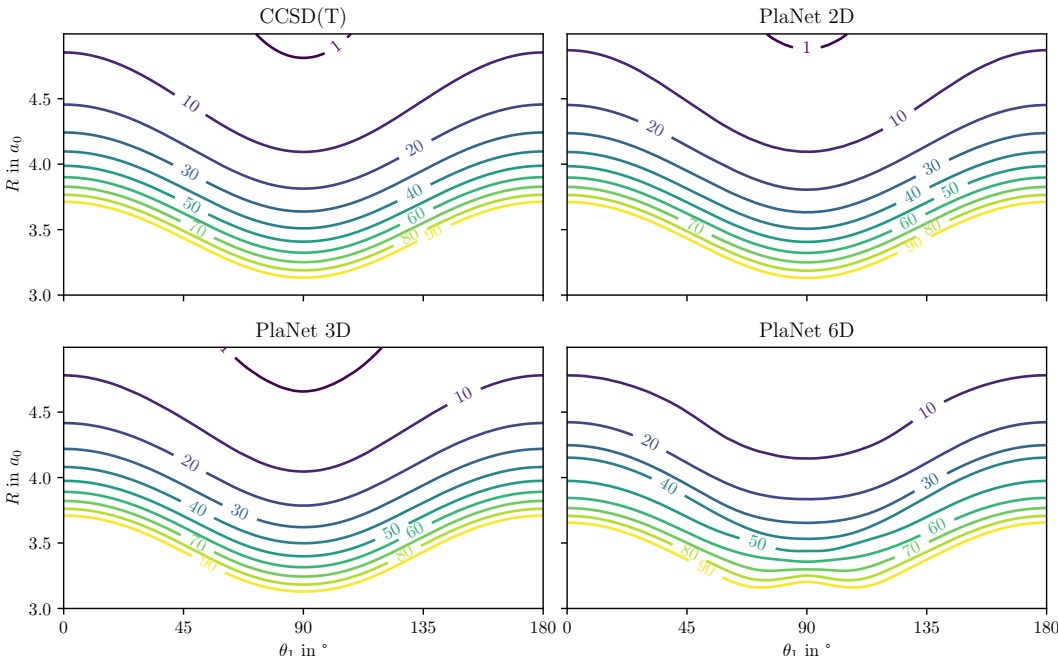

Figure 12: Comparison between different PlaNets trained on a two-dimensional slide (top right), three-dimensional slice (bottom left), and the full six-dimensional energy surface of $H_2$-HF compared to the reference CCSD(T) energy surface (Yang et al., 2018).

Gauche conformer as well as the two transition states TS1 and TS2. All energy surfaces are plotted in Figure 11. Note that while the reference calculations for Gauche in Figure 11a are CCSD(T) based, for both transition states we use reference data from an $\Delta$-ML approach (Nandi et al., 2022). While all surfaces generally agree on the approximate shape of the energy surface, we notably identify differences between our PlaNet estimates and the reference data for the Gauche and TS1 states. For the Gauche conformer, we locate a slightly different minimum and for TS1 we find the $120°$ periodicity to be broken in the given geometries. Note that the broken symmetry is due to working on partially relaxed structures (Nandi et al., 2022).

In terms of relative error, we observe a maximum discrepancy of $0.63\,\mathrm{m}E_\mathrm{h}$, $0.55\,\mathrm{m}E_\mathrm{h}$ and $0.21\,\mathrm{m}E_\mathrm{h}$ for Gauche, TS1, and TS2, respectively.

### G.5 HIGHER DIMENSIONAL ENERGY SURFACES

While the main body already contains results on the two-dimensional energy surface of $H_2$-HF, we are often interested in the full-dimensional energy surface. Though, obtaining sufficient samples from higher dimensional surfaces is a challenge due to the curse of dimensionality. To see how well our PlaNet framework generalizes to more dimensions, we train additional models on a three-dimensional slice and the full six-dimensional energy surface of $H_2$-HF. For the three-dimensional training, we additionally include the $\theta_2$ angle. Albeit training on additional dimensions, we stick to the comparison of the same two-dimensional slice as in the main body to ensure comparability.

Figure 12 shows a comparison between the CCSD(T) baseline, and the three differently trained PlaNet models. While the general shape is similar between all models, it is noticeable that the additional dimensions cause the energy surface to deform at angles of $\theta_1 \approx 90°$ increasing the MAE from $0.28(9)\,\mathrm{m}E_\mathrm{h}$ to $1.02(34)\,\mathrm{m}E_\mathrm{h}$ and $2.39(25)\,\mathrm{m}E_\mathrm{h}$, respectively. We hypothesize that this is due to the curse of dimensionality and the consequent scarcity of data. Note that while the CCSD(T) baseline required first the selection and evaluation of 35.000 individual data points and an additional function fit, our PlaNet framework is able to approximately capture the energy surface with a single training. Although we do not know the exact time requirements for the CCSD(T) calculations, we expect them to be in the range of a few minutes per geometry on a modern CPU, i.e., if we take an

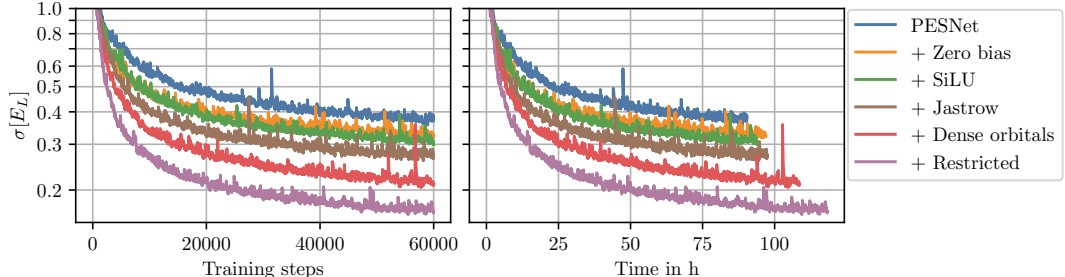

Figure 13: Training plots on the nitrogen dimer with different enhancements to PESNet. On the left, the standard deviation of the local energy (lower is better) is plotted on the y-axis, and the training steps on the x-axis. The right plot shares the same y-axis but uses the training time ax x-axis. It is apparent that all enhancements of PESNet++ strictly enhance the efficiency of PESNet as they provide lower errors in less time and steps.

Table 6: Ablation study on the number of surrogate training steps $N_{\text{surr}}$.

| $N_{\text{surr}}$ | $H_4^+$ | $H_4$ | $Li_2$ | $H_{10}$ | $N_2$ | $H_2$-HF |
|---|---|---|---|---|---|---|
| 1 | 0.0124(4) | 0.0084(8) | 0.0819(13) | 0.14(4) | 0.57(21) | 0.338(31) |
| 5 | 0.0117(4) | 0.0072(7) | 0.0786(9) | 0.14(7) | 0.33(7) | 0.333(35) |
| 10 | 0.0115(4) | 0.00551(20) | 0.0758(12) | 0.19(8) | 0.32(6) | 0.33(4) |

optimistic guess of 1 minute per evaluation, the total compute time for the energy surface is at least 24 days. In contrast, our PlaNet model has been trained in only 4.5 days with a single Nvidia A100 GPU.

## H CONVERGENCE

To analyze the efficiency of our proposed enhancements to PESNet, we investigate the convergence rate based on terms of the number of steps as well as time. The resulting training curves for the nitrogen molecule are plotted in Figure 13. it can be seen that all enhancements strictly improve convergence and result in significantly lower errors. While especially the dense orbitals add a non-trivial amount of additional computational requirement, they also result in significantly better energies. The convergence rate per time and per step is strictly improving for all of our proposed improvements.

## I ABLATION STUDIES

To evaluate the value of our training heuristics, we perform ablation studies on the number of surrogate training steps $N_{\text{surr}}$ and the exponential moving average decay rate $\gamma$. For the first experiment, we alter the $N_{\text{surr}}$ hyperparameter between 1, 5, and 10 and run each experiment 5 times. The results are listed in Table 6. It can be seen that performing more than one step typically improves the results across all systems. This effect is more pronounced for larger systems.

Table 7: Ablation study with fixed values for the decay $\gamma$ of the exponential moving average.

| $\gamma$ | $H_4^+$ | $H_4$ | $Li_2$ | $H_{10}$ | $N_2$ | $H_2$-HF |
|---|---|---|---|---|---|---|
| 0.99 | 0.0117(4) | 0.0076(6) | 0.0786(9) | 0.112(21) | 0.57(15) | 0.39(4) |
| 0.9999 | 0.019(4) | 0.167(30) | 0.087(4) | 0.55(7) | 0.46(14) | 0.26(5) |
| adaptive | 0.0117(4) | 0.0072(7) | 0.0786(9) | 0.14(7) | 0.33(7) | 0.333(35) |

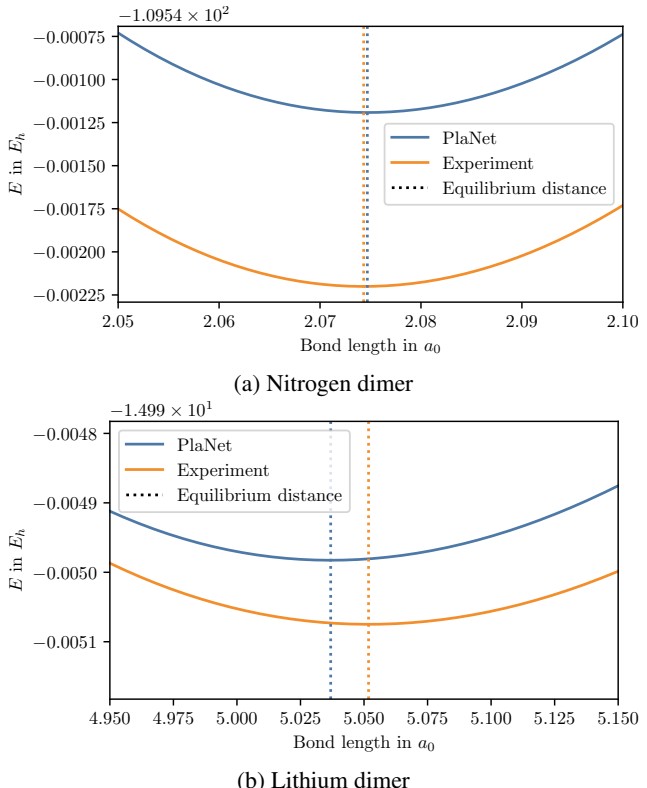

(a) Nitrogen dimer

(b) Lithium dimer

Figure 14: High-resolution energy surface scans around the equilibrium structure.

In the second experiment, we fix the exponential moving average parameter $\gamma$ instead of adapting it. The results in Table 7 show that an initial high decay factor results in significantly higher final errors due to the large influence of the initial bad labels. While a low $\gamma$ works reasonably across all systems, we are able to obtain better results on larger systems with our adaptive scheduling.

In general, we find our PlaNet framework to be robust against different hyperparameter settings.

## J SYSTEMS

In the following, we list all geometries used throughout this paper:

- Hydrogen rectangle $H_4$: geometries taken from Pfau et al. (2020).
- Lithium dimer $Li_2$: 32 evenly distributed distances between $3.5\,a_0$ and $14\,a_0$.
- Hydrogen chain $H_{10}$: geometries taken from Motta et al. (2017).
- Nitrogen dimer $N_2$: geometries taken from Pfau et al. (2020).
- $H_2$-HF: 64 regular grid points of the N-dimensional energy surface. The boundaries are chosen as: $r_1, r_2 \in [1.2\,a_0, 1.8\,a_0]$, $R \in [3.0\,a_0, 8.0\,a_0]$, $\theta_1, \theta_2, \phi \in [0°, 180°]$.
- Ethanol $C_2H_5OH$: 64 evenly distributed torsion angles between $0°$ and $360°$.

## K EQUILIBRIUM STRUCTURES

To find equilibrium structures, we performed high-resolution energy surface scans to find the global minimum. As many structures result in identical energies when performing inference with float32 precision, we switch to float64 precision. Note that all models were still trained with float32 and just converted for finding the equilibrium structure. We visualize these high dimensional energy surfaces around the equilibrium geometries for the nitrogen and lithium dimer in Figure 14.

