# OpenReview forum: "Sampling-free Inference for Ab-Initio Potential Energy Surface Networks"
_ICLR.cc/2023/Conference — ICLR 2023 poster_

### Official Review · Reviewer_jX7P · 2022-10-20

**Confidence:** 3
**Clarity, Quality, Novelty And Reproducibility:** This paper looks good in clarity, qua…
**Correctness:** 4
**Technical Novelty And Significance:** 3
**Empirical Novelty And Significance:** 3
**Recommendation:** 8

**Strength And Weaknesses:**

Strengths: 1. PESNET++ can produce highly accurate energy results (compared to FermiNet).
                  2. PlaNet can approximate PESNET++ results well, but avoid the QMC inference phase, and thus can produce PES results much more efficiently.

Weakness: The model is still limited, not applicable for real-world PES, for instance, molecular dynamics.

Some questions to the authors: 1. Does PlaNet still have the variational property? 2. The neural Jastrow factor is not considering cusp condition as the traditional ones. Is it by design or just for convenience?

**Summary Of The Paper:**

This paper proposed two improvements over the baseline model PESNet. The first one, PlaNet, use the local energy computed in each PESNET training step to train a surrogate model, which takes in the geometry of nucleus and output final energy, namely no dependence on electrons. At inference time, only the surrogate model is used to do prediction. Hence, the orders of magnitude for inference is faster than PESNet. The second one, PES++, includes several improvements to achieve better accuracy than PES with similar cost.

**Summary Of The Review:**

This paper proposed two improvements over PESNet, which makes the inference of PESNet faster and more accurate. The paper is written in a good manner.

---

> ### Author Response · Authors · 2022-11-11
> **Official Response to Reviewer jX7P**
>
> Thank you for your effort in reviewing our work, we sincerely appreciate that you find our work being a significant improvement over previous work.
>
> We are happy to answer your questions:
>
> > **Does PlaNet still have the variational property?**
>
> Unfortunately, PlaNet does not provide a variational guarantee, i.e., a badly trained surrogate model may result in lower than ground-state energies. Though, in our empirical evaluation, we did not find this to be a significant concern due to the close relation to the via VMC obtained results. On a side note, while the variational property is desirable, it is not essential. Many of the commonly used methods do not provide variational bounds, e.g., density functional theory (DFT) or coupled-cluster theory (CC).
>
> ---
> > **The neural Jastrow factor is not considering cusp condition as the traditional ones. Is it by design or just for convenience?**
>
> While we initially experimented with Jastrow factors that enforce the cusp condition as they are typically found in computational chemistry literature or the PauliNet [1], we did not find their performance to be compelling compared to the neural Jastrow factor. In the FermiNet paper [2], the authors show that a network even without any Jastrow factor can learn correct cusp conditions if the euclidean norm is supplied as input to the network. This evidence is further supported by [3] where the authors show that a periodic equivalent of the euclidean norm is sufficient to learn the cusp conditions in periodic systems correctly.
>
> [1] "Deep-neural-network solution of the electronic Schrödinger equation" by Hermann et al., 2020
>
> [2] "Ab initio solution of the many-electron Schrödinger equation with deep neural networks" by Pfau et al., 2020
>
> [3] "Discovering Quantum Phase Transitions with Fermionic Neural Networks" by Cassella et al., 2022

---

### Official Review · Reviewer_oynR · 2022-10-21

**Confidence:** 4
**Correctness:** 4
**Technical Novelty And Significance:** 3
**Empirical Novelty And Significance:** 3
**Recommendation:** 8

**Clarity, Quality, Novelty And Reproducibility:**

The paper is clear, technically sound, and of sufficient novelty to be of very high interest to the ML subcommunity interested in physical chemistry.

**Strength And Weaknesses:**

The paper is well written and nicely presented.

The idea of using an energy surrogate as part of the VMC is promising.

The proposed improvements to PESNet seem relevant and improve performance.

Weaknesses:
The training of the surrogate energy model is done via samples (targets) which only asymptotically are correct, which is handled by training only on the newest samples. Because of label noise, a heuristic exponential moving average is also used. It could be interesting with some further theoretical insight into the convergence properties of this approach.

I expect that most of the paper might be difficult to follow for the general ML audience.


**Summary Of The Paper:**

The paper proposes a method for speeding up the learning of potential energy surfaces by improving PESNet (a neural wavefunction model parameterized by a graph neural network and trained by variational Monte Carlo). In each step, previous methods run Monte Carlo simulations to estimate the energy, and the paper proposes to use these estimated energies to train an energy surrogate GNN (a Dimenet++ model). In addition, several changes to the PESNet neural wave function are presented.

**Summary Of The Review:**

An interesting and well written paper that takes a small but important step towards solving the Schrodinger equations in a more efficiently and precisely via ML.

---

> ### Author Response · Authors · 2022-11-11
> **Official Response to Reviewer oynR**
>
> We thank the reviewer for his effort in reviewing our work and highly appreciate that he/she finds our work an important step in solving the Schrödinger equation efficiently.
>
> > **It could be interesting with some further theoretical insight into the convergence properties of this approach.**
>
> In variational Monte Carlo, averaging energies either via fixed windows or exponential moving averages are typical to estimate the final energy of a molecular system.
> Since we model the PES as a whole, we have to average functions rather than scalars.
> Unfortunately, in function space, this would be equivalent to averaging in the codomain of the function which is non-trivial for arbitrary functions.
> We instead average in the parameter space of the function.
> As the gap between parameter sets decreases, this approximation is getting closer to the average in the codomain of the function as the function can be represented by its first-order Taylor approximation, i.e., a linear function.
> As the parameter sets are obtained via subsequent gradient descent steps, the distance between parameter sets is small in our setting and the approximation error small.
>
> While this is no theoretical convergence analysis, it is a strong motivation for averaging in parameter space.
>
> ---
> > **I expect that most of the paper might be difficult to follow for the general ML audience.**
>
> We understand that quantum chemistry is not a typical topic for ML conferences and includes a lot of unfamiliar terminologies. We tried our best to serve the ML audience as well as the quantum chemistry reader. If there are any unclear sections/paragraphs, we would be very grateful if you could point them out to us.

---

### Official Review · Reviewer_CnTt · 2022-10-23

**Confidence:** 4
**Correctness:** 3
**Technical Novelty And Significance:** 3
**Empirical Novelty And Significance:** 4
**Recommendation:** 6

**Clarity, Quality, Novelty And Reproducibility:**

- The writing of this paper is overall clear and high-quality. However, it combines two slightly independent contributions (PlaNet and PESNet++) into a single paper. The narration could be improved to better clarify the connection between the two contributions.
- The paper introduces multiple innovations for both the PlatNet and PESNet++ to improve the PESNet architecture. The novelty of individual innovations are not very high, but they are combined to solve the important challenge of VMC and achieve the SOTA in several molecules.
- The source code is not provided for review. I presume that the code will be provided possibly upon acceptance, because the paper says code will be open.


**Strength And Weaknesses:**

Strength:
- PlatNet introduces an online learning scheme that learns a PES using the noisy labels from the VMC optimization. This innovation is significant because it enables the use of the very noisy energy data from VMC.
- Multiple innovations are introduced to improve the PESNet++, which results in a new SOTA in VMC for several molecules.

Weakness:
- The motivation of learning a surrogate PlaNet from VMC optimization is unclear. From my understanding, the authors first use VMC to explore the entire PES and use the noisy data generated in the process to train a surrogate PlatNet. Since the entire PES has already been explored with VMC, why do we need a surrogate PatNet to predict the PES again? One argument may be that PatNet can generalize to new configurations that are not simulated by VMC. In that case, the authors probably should compare with a baseline DimeNet++ that is trained on the clean PES data from VMC to demonstrate the advantage of learning from noisy VMC energy.
- The generalization performance of PlaNet is not studied in this paper. In Figure 3, 4, the PESNet++ simulated configurations are extremely dense and PlatNet only interpolates between nearby points.
The discussion on the results in Table 3 is slightly misleading. It is unfair to compare the training/inference time of the surrogate model to VMC, because the latter requires the former to generate data.


**Summary Of The Paper:**

This paper aims to simulate the potential energy surfaces of atomic systems by accelerating the variational Monte-Carlo (VMC) approach. It builds on the PESNet by Gao and Gunnemann and contributes two major innovations: 1) it introduces PlaNet which trains a surrogate model to avoid the expensive Monte-Carlo integration; 2) it introduces PESNet++ which improves multiple aspects of the PESNet. It shows 7 orders of magnitude acceleration and reduces the error up to 74%.

**Summary Of The Review:**

In summary, the paper makes two contributions: 1) a PlaNet that learns a PES from the noisy data of VMC; 2) a PESNet++ that improves PESNet and achieves SOTA in several molecules. I think the motivation of 1) needs to be further clarified and additional baseline may be required. Contribution 2) alone makes significant progress in the ab-initio modeling of molecules. The paper may be accepted based on contribution 2) but the paper needs to be significantly revised. I give a score of 5 for now, but I am open to increase my score after satisfying responses from the authors.

---

> ### Author Response · Authors · 2022-11-11
> **Official Response to Reviewer CnTt**
>
> We thank you for your detailed review and highly appreciate that you find PESNet++ to be a significant progress in ab-initio methods.
>
> Unfortunately, we believe that there might be a misunderstanding regarding the problem setting of PlaNet. We hope to resolve this and improve our manuscript to avoid future misconceptions.
>
> > **Since the entire PES has already been explored with VMC, why do we need a surrogate PatNet to predict the PES again?**
>
> **Short answer:** After training PESNet++, we obtain a neural wave function for all geometries but we do not have the associated energies. PlaNet directly provides the energy surface skipping the typically required numerical integration to get from the wave function to the energy.
>
> **Long answer:** During VMC training, PESNet++ learns on a set of changing geometries sampled from some continuous domain of geometries. During training, we have to compute approximate energies to obtain gradient updates but as we use very small batch sizes per geometry (e.g., 64) these are very noisy. Further, as we resample geometries from a continuous space each step, we will never see the same geometry twice prohibiting simply averaging these noisy energies. Thus, after training, we obtain a neural wave function that generalizes over the domain of geometries, but no energy surface. To obtain the energy of a specific geometry, we still have to plug the wave function into the Schrödinger equation and solve for the energy. This is done via Monte Carlo integration with $O(10^6)$ samples. This process is very slow especially if it has to be done for many geometries, e.g., to model an energy surface. PlaNet addresses this by systematically reusing the noisy energies which were computed during training. Thus, to train the surrogate we never need to perform any additional MC integration. For our figures, we additionally performed the expensive MC integration for comparison but the surrogate has not been trained with these.
>
> We update the second paragraph in our introduction:
> > *While training significantly faster, afterward one only obtains a neural wave function that generalizes over a domain of geometries, but not the associated energy surface.*
>
> ---
> > **the authors probably should compare with a baseline DimeNet++ that is trained on the clean PES [...].**
>
> This would be the traditional method of obtaining the continuous energy surface. We did not perform such a comparison as it is very similar to directly comparing to the clean PES that we include. The main difference is that the traditional method requires significantly more compute time, e.g., for ethanol, this would require additional $\approx2.7h*64=172.8h$ just for generating the clean PES. Meanwhile, PlaNet does not require any additional inference as we simply use the noisy labels obtained during training.
>
> We do not expect a significant benefit of PlaNet over the traditional method in terms of accuracy but find our method to be significantly faster.
>
> ---
> > **The generalization [...] is not studied in this paper.**
>
> As explained in our first answer, we believe that generalization is no concern as PlaNet's goal is to accelerate the inference of PESNet++. If one were interested in additional geometries, extending the training domain of PESNet++ accomplishes this generalization with likely better results.
>
> ---
> > **It is unfair to compare the training/inference time of the surrogate model to VMC [...].**
>
> With our first comment in mind, we hope that our discussion on training and inference times is sound.
> Still, we would like to use this opportunity to discuss our reasoning.
>
> In section 6.3, we outline two key points:
> 1) Surrogate training is cheap compared to VMC training and in the same order of magnitude as a single VMC inference, i.e., it does not add a lot to the overall compute time.
> 2) Surrogate inference is extremely fast compared to VMC inference.
>
> As the surrogate inference is an **alternative** to VMC inference with no dependency, we believe that comparing their runtimes is adequate. Though, for a complete picture, one must also factor in the additional training time of the surrogate. But, as the number of geometries varies, a total inference time seems unsuitable, e.g., for a single geometry PlaNet is approximately as fast as VMC integration but if we choose $10^6$ geometries for our energy surface, PlaNet is $\approx 10^6$ times faster. Thus, we decided to split the timings into training and inference/geometry rather than reporting a total inference time.
>
> If you still find this comparison unfair, we are happy to discuss this issue to improve the presentation of our work.
>
> ---
> > **The source code is not provided for review.**
>
> We are sorry for the inconvenience, we intended on sharing the code directly with the reviewer as recommended by the [author guide](https://iclr.cc/Conferences/2023/AuthorGuide). Unfortunately, this was not possible until now. We provided a link to our source code in a general comment to all reviewers.

---

> > ### Comment · Reviewer_CnTt · 2022-11-19
> > **Thank you for you reply!**
> >
> > I appreciated the authors for their detailed reply. The comments have addressed a significant part of my concern. My remaining question is whether it is possible to use significantly less steps than $O(10^6)$ in the Monte Carlo integration to generate noisy labels, and use a DimeNet++ to learn a clean PES from the noisy data. It will be similar to the PlaNet idea, but only uses the trained PESNet(++) to generate data.
> >
> > Overall, I still think this work is a valuable contribution. I will also update my score. But the authors might consider to tune down the PlaNet part a little. I believe there may be more efficient approaches to learn a surrogate model besides reusing intermediate noisy data from the training of PESNet(++).
> >
> > Finally, since the end goal is to speed up Monte-Carlo integration, "accelerates inference by 7 orders of magnitude" might be an over-claim, because the training time of the PlaNet is not counted.

---

> > > ### Author Response · Authors · 2022-11-19
> > > **Training with samples from a trained PESNet++**
> > >
> > > Thank you for your response!
> > >
> > > We agree that this is an interesting point of comparison. So, we ran a few experiments on ethanol and the nitrogen dimer to judge how such a "post-hoc" trained DimeNet++ would perform. For ethanol, we plotted both the absolute energy surface ([figshare](https://figshare.com/s/6f260057998bc771be18?file=38258082)) as well as the relative ([figshare](https://figshare.com/s/6f260057998bc771be18?file=38258085)) and for nitrogen the usual relative energy surface([figshare](https://figshare.com/s/6f260057998bc771be18?file=38258091)).
> > >
> > > These plots require some explanation: The PlaNet, PESNet++, experimental and CCSD(T) energies are the same as in the paper. The other lines indicate post-hoc trained DimeNet++ on energies obtained from PESNet++ **after** training. The lines differ in the total number of electron configurations (distributed across the geometries) we used to estimate the target energies. For instance, for ethanol "1e6 samples" means 1.000.000 electron configuration distributed over 64 geometries, i.e., 15.625 samples per geometry. For nitrogen, the same label results in 26.315 samples per geometry due to having fewer reference geometries 38 vs 64. Note that the sets are subsets of each other, i.e., $ie6 \text{ samples}\subset je6\text{ samples}$ if $i<j$.
> > >
> > > With this description in mind, both plots suggest that PlaNet works approximately as well as having 16e6 to 32e6 electron configurations distributed over the energy surface. But, in contrast to these post-hoc methods, PlaNet did not require any additional inference. The following two tables list the additional approximate cost of generating these post-hoc datasets for both molecules.
> > >
> > > Tables of additional compute time to generate the datasets:
> > > ### Ethanol
> > >
> > > |      |   1e6 samples |   2e6 samples |   4e6 samples |   8e6 samples |   16e6 samples |   32e6 samples |
> > > |:-----|--------------:|--------------:|--------------:|--------------:|---------------:|---------------:|
> > > | Time in h |   2.7 |     5.4  |       10.7 |      21.4  |      42.9   |      85.7   |
> > >
> > >
> > > ### Nitrogen
> > >
> > > |      |   1e6 samples |   2e6 samples |   4e6 samples |   8e6 samples |   16e6 samples |   32e6 samples |
> > > |:-----|--------------:|--------------:|--------------:|--------------:|---------------:|---------------:|
> > > | Time in h |   0.9 |      1.7  |       3.5 |      7.0  |      13.9   |      27.9   |

---

> ### Author Response · Authors · 2022-11-18
> **Gentle Reminder**
>
> In light of the end of the author-reviewer discussion period today, we would again like to highlight our response. We hope that we resolved the misconception about the motivation behind PlaNet and addressed your concerns adequately. If there are any outstanding or further questions we are delighted to discuss these.

---

### Official Review · Reviewer_7yt6 · 2022-10-25

**Confidence:** 3
**Correctness:** 3
**Technical Novelty And Significance:** 3
**Empirical Novelty And Significance:** 3
**Recommendation:** 5

**Clarity, Quality, Novelty And Reproducibility:**

What is Eref in Figure 3 ? r12-MR-ACPF would seem to be working great if it wasn't for some systematic bias.

" is in perfect agreement with the experimental results(Le Royet al., 2006) about the equilibrium structure at 2.074 a0." It is not clear by visual inspection where the minima is, the points seem to move randomly up and down from the true surface

The stoichiometric indices of chemicals should be subscripted

"PlaNet almost perfectly recovers the CCSD(T) results with a maximum disagreement of 0.13 mEh while only requiring ≈ 0.15 ms"  Does this factor in training time ?

What phase space does the PESNet algorithm explore?

The Energies reported are for the equilibrium geometry ? (Table 1, 2, 5 )

**Details Of Ethics Concerns:**

"we will release our source code under the Hippocratic" the Hippocratic what?

The quantum chemistry for chemical weapons is too far fetched to even deserve including

**Strength And Weaknesses:**

Strengths

The approach seems fundamentally sound and has good performance. The surrogate model is rather accurate and accelerates inference with little loss of accuracy, whereas the approximations and model updates based on traditional quantum chemistry heuristics are gainful and results in good cost/accuracy tradeoffs.

Weaknesses.

Neither of the innovation of performance improvements are particularly innovative as far as ML is concerned and this seems more fit for a quantum chemistry scientific journal. The use of a surrogate force-field-like potential seems relatively trivial and does not present a large methodological or task-specific innovation. All the quantum chemistry tricks are effective, but again, not that innovative on the ML side.

**Summary Of The Paper:**

This paper builds on the PESNet model of NN-based wave functions for small polyatomic molecules [this reviewers recommends avoiding re-using names for model architectures to avoid compromising anonymity). The works proposes two innovations. One is learning a surrogate model of the VMC energies to avoid O(N^4) scaling in the MC integration of the VMC integration. The other is a series of updates based on common approximations and tricks in quantum mechanics

**Summary Of The Review:**

This paper reports exciting practical improvements to an ML-based model for generatic wave functions and energies of small molecules. The method seems to provide desired improvements in speed while retaining accuracy, but the innovations do not seem to be arising from ML insights and are mostly through use of a surrogate function for energies, which is a common strategy in chemistry and from inclusion of quantum mechanical approximations that are well stablished.

---

> ### Author Response · Authors · 2022-11-11
> **Official Response to Reviewer 7yt6 - Part 1**
>
> Thank you for your detailed review, we highly appreciate that you find our work sound and well-performing.
>
> In the following, we hope to address your concerns adequately:
>
> > **Neither of the innovation of performance improvements are particularly innovative as far as ML is concerned [...].**
>
> We kindly disagree with this criticism.
> Ignoring the application to quantum chemistry, PlaNet essentially tackles the challenge of "How can we learn a temporally evolving function from noisy observations?". To solve this problem, we find three key steps in solving such problems:
> 1) By learning in an online fashion we can evolve our function representation jointly with the underlying data.
> 2) An exponential moving average on the parameters of the function model suffices as a proxy to averaging in the codomain of the function.
> 3) By comparing the error distribution to the data distribution we developed an adaptive decay scheme that results in a good balance between bias and variance compared to fixed decay rates.
>
> Beyond these general insights, we strongly believe that our work fits very well in the subject areas listed in the ICLR 2023 [call for papers](https://iclr.cc/Conferences/2023/CallForPapers) which specifically include applications of machine learning to sciences during the submission phase. As you agree with us, our work significantly advances state-of-the-art neural network wave functions for quantum chemistry.
>
> To accomplish this, we incorporate both insights from machine learning as well as quantum mechanics to improve our method, e.g., we adopt a neural network Jastrow factor combining ML with a QM design and translate the doubly occupied orbitals from restricted Hartree-Fock theory to neural network wave functions.
>
> ---
>
> > **$E_\text{ref}$ what is the reference energy? r12-MR-ACPF would seem to be working great if it wasn’t for some systematic bias.**
>
> As standard in the literature [1,2,3], we use the experimental dissociation energies from [4] plus twice the ground-state energy of the nitrogen atom taken from [5]. You are definitely right, r12-MR-ACPF has very compelling results, thus, the inclusion in our comparison. However, compared to neural wave functions, r12-MR-ACPF scales factorially, i.e., $O(N!)$, compared to the typical neural wave function scaling of $O(N^4)$.
>
> To better clarify this in our paper, we changed the y-label from $E-E_\text{ref}$ to $E-E_\text{experiment}$.
>
> ---
> > **It is not clear by visual inspection where the minima is, the points seem to move randomly up and down from the true surface.**
>
> This is due to plotting energy differences. To obtain the equilibrium distance, we generated a high-resolution scan around the equilibrium and searched for the global minimum. To better illustrate this, we plot the absolute energy surface of PlaNet and of the experimental results closely around the equilibrium in this [figure on figshare](https://figshare.com/s/6f260057998bc771be18). In creating this plot, we noticed that float32 precision is insufficient for finding minima and, thus, perform inference in float64 (the model has still been trained with float32 precision). Due to the higher precision, we find slightly different values for the equilibrium distance than in our initial submission: $2.0747a_0$ and $2.0743a_0$ for PlaNet and the experimental results [4], respectively. Thank you for pushing us to take a closer look at this, we updated the manuscript accordingly.
>
> Moreover, we added Appendix K discussing how we find equilibrium structures where we also include the figure from figshare and a similar one for the lithium dimer.
>
> ---
> > **"PlaNet almost perfectly recovers the CCSD(T) results with a maximum disagreement of 0.13 mEh while only requiring ≈ 0.15 ms" Does this factor in training time ?**
>
> These timings only refer to pure inference time per geometry. It does not include VMC training nor surrogate training. The total time to obtain the energies for the 64 VMC energies can be computed as $478h \text{ VMC training} + 64\times 2.7h \text{ VMC inference}=650.8h \text{ total time}$. For PlaNet the calculation is $478h \text{ VMC training} + 0.72h \text{ surrogate training} + 64\times 0.1ms \text{ surrogate inference}=478.72h \text{ total time}.$
>
>
> To be more precise in our wording, we updated the phrasing:
> >  *PlaNet almost perfectly recovers the CCSD(T) results with a maximum disagreement of 0.13 mEh while only requiring $\approx 0.11$ms per inference after training compared to the $\approx 2.7$h per inference required for VMC integration.*
>
> Note that we corrected the inference time to 0.11ms, the 0.15ms were accidentally measured on different hardware.

---

> > ### Author Response · Authors · 2022-11-11
> > **Official Response to Reviewer 7yt6 - Part 2**
> >
> > > **The Energies reported are for the equilibrium geometry? (Table 1, 2, 5 )**
> >
> > The energies in Tables 1, 2, and 5 are all averaged over whole energy surfaces. Specifically:
> > * Hydrogen rectangle H$_4$: geometries taken from [1,2].
> > * Lithium dimer Li$_2$: 32 evenly distributed distances between $3.5a_0$ and $14a_0$.
> > * Hydrogen chain H$_{10}$: geometries taken from [6].
> > * Nitrogen dimer N$_2$: geometries taken from [1,2].
> > * H$_2$-HF: 64 regular grid points of the two-dimensional energy surface.
> > * Ethanol C$_2$H$_5$OH: 64 evenly distributed torsion angles between 0° and 360°.
> >
> > We added this list in the new Appendix J and added updated the paragraph in the main body:
> > > *Lastly, we analyze PlaNet's reconstruction of VMC energies across a variety of energy surfaces, see Appendix J for the list of used geometries.*
> >
> > ---
> > > **What phase space does the PESNet algorithm explore?**
> >
> > Like [2] and as stated in the introduction and the background on variational Monte Carlo, we are only concerned with the ground-state wave function/energy:
> > > To obtain the ground-state energy for a fixed molecular system, one has to solve the associated time-independent Schrödinger equation. [...] In linear algebra, Equation (1) is an eigenvalue problem where we are interested in the lowest eigenvalue $E_0$ which is also called the ground-state energy.
> >
> > ---
> > > **The stoichiometric indices of chemicals should be subscripted**
> >
> > Thank you for the suggestion, we adopted the change throughout the manuscript.
> >
> > ---
> > > **"we will release our source code under the Hippocratic" the Hippocratic what?**
> >
> > Thank you for catching this typo, we meant the Hippocratic license.
> >
> > Also, we released the source code to the reviewers in a general comment and will publicly share it upon acceptance.
> >
> > ---
> >
> > [1] "Ab initio solution of the many-electron Schrödinger equation with deep neural networks" by Pfau et al., 2020
> >
> > [2] "Ab-Initio Potential Energy Surfaces by Pairing GNNs with Neural Wave Functions" by Gao et al., 2022
> >
> > [3] "Gold-standard solutions to the Schrödinger equation using deep learning: How much physics do we need?" by Gerard et al, 2022
> >
> > [4] "An accurate analytic potential function for groundstate N2 from a direct-potential-fit analysis of spectroscopic data." by Le Roy et al., 2006
> >
> > [5] "Ground-state correlation energies for atomic ions with 3 to 18 electrons" by Chakravorty et al., 1993
> >
> > [6] "Towards the Solution of the Many-Electron Problem in Real Materials: Equation of State of the Hydrogen Chain with State-of-the-Art Many-Body Methods" by Motta et al., 2017

---

> > > ### Comment · Reviewer_7yt6 · 2022-11-19
> > > **Acknoweledged**
> > >
> > > I have read and understood the authors' reply. There are not technical issues that i observe, and the method is definitely novel, complex and solves a hard problem. I am just having trouble judging its importance and implications. The replies did address my concerns so i am updating my score slightly.

---

> ### Author Response · Authors · 2022-11-18
> **Gentle Reminder**
>
> In light of the end of the author-reviewer discussion period today, we would again like to highlight our response and the manuscript's revised sections. We hope that we adequately address your concerns and are interested in your thoughts. If there are any outstanding or further questions we are delighted to discuss these.

---

### Decision · Program_Chairs · 2023-01-20

**Decision:**

Accept: poster

**Justification For Why Not Higher Score:**

This paper is heavily quantum chemistry flavor. Its audience will be relatively narrow. We do not think positioning it as a spotlight or oral paper makes too much sense.

**Justification For Why Not Lower Score:**

Most reviewers like this paper, and most concerns have been addressed by the author rebuttal.

**Metareview: Summary, Strengths And Weaknesses:**

This paper aims to simulate the potential energy surfaces of atomic systems by accelerating the variational Monte-Carlo (VMC) approach.  It introduces PlaNet which trains a surrogate model to avoid the expensive Monte-Carlo integration, and introduces PESNet++ which improves multiple aspects of the PESNet. It shows several orders of magnitude acceleration and reduces the error significantly at the same time.

Overall speaking, this is a nice paper and most reviewers like it. One concern raised during the review process is the technical novelty in terms of machine learning, while the reviewer acknowledged that all the quantum chemistry tricks are effective. The authors had several rounds of discussions and successfully convinced the reviewer about the technical novelty of the paper. Given this, we believe this paper has sufficient value to be accepted.


**Note From Pc:**

if the above contains the word "oral" or "spotlight" please see: "oral" presentation means -> notable-top-5% and "spotlight" means -> notable-top-25%. As stated in our emails, we are disassociating presentation type from AC recommendations